# Genetic Insights into Breast Cancer in Northeastern Mexico: Unveiling Gene–Environment Interactions and Their Links to Obesity and Metabolic Diseases

**DOI:** 10.3390/cancers17060982

**Published:** 2025-03-14

**Authors:** Hugo Leonid Gallardo-Blanco, María de Lourdes Garza-Rodríguez, Diana Cristina Pérez-Ibave, Carlos Horacio Burciaga-Flores, Víctor Michael Salinas-Torres, Moisés González-Escamilla, Rafael Piñeiro-Retif, Ricardo M. Cerda-Flores, Oscar Vidal-Gutiérrez, Celia N. Sanchez-Dominguez

**Affiliations:** 1Servicio de Oncología, Centro Universitario Contra el Cáncer (CUCC), Hospital Universitario “Dr. José Eleuterio González”, Universidad Autónoma de Nuevo León, Monterrey 66451, NL, Mexico; hugo.gallardobl@uanl.edu.mx (H.L.G.-B.); maria.garzarg@uanl.edu.mx (M.d.L.G.-R.); dperezi@uanl.edu.mx (D.C.P.-I.); bufc1655389@uanl.edu.mx (C.H.B.-F.); drmoisesgzz87@gmail.com (M.G.-E.); rpineiror@uanl.edu.mx (R.P.-R.); oscar.vidalgtr@uanl.edu.mx (O.V.-G.); 2Departamento de Medicina Genómica, Hospital General Culiacán “Dr. Bernardo J. Gastélum”, Servicios de Salud del Instituto Mexicano del Seguro Social para el Bienestar, Culiacán 80064, SIN, Mexico; vm_salinas7@hotmail.com; 3Facultad de Medicina, Universidad Autónoma de Sinaloa, Culiacán 80019, SIN, Mexico; 4Facultad de Enfermería, Universidad Autónoma de Nuevo León, Monterrey 64460, NL, Mexico; ricardocerda_mx@yahoo.com.mx; 5Departamento de Bioquímica y Medicina Molecular, Facultad de Medicina, Universidad Autónoma de Nuevo León, Monterrey 64460, NL, Mexico

**Keywords:** breast cancer, genetic risk variants, obesity, overweight, insulin resistance, inflammation, menopause, gene–environment interactions, *MMP8*, *FTO*

## Abstract

Breast cancer is a common disease, and its occurrence in Mexico has been rising over the last decade. This study investigates specific genetic DNA variants in women from Northeastern Mexico to understand how genes and age, body mass index, and menopause contribute to breast cancer risk. Studying interactions between gene variants related to insulin resistance, obesity, and inflammation may clarify breast cancer development. This knowledge improves our understanding of breast cancer risk factors and finding biomarkers for personalized prevention strategies in Mexican mestizo women.

## 1. Introduction

Breast cancer (BC) is the most frequent cancer in women worldwide (incidence), with 2,295,686 new cases and 665,684 deaths in women in 2022 [1]. BC represents 23.8% of the total new cases of cancer diagnosed in women. In Mexico, there were 31,043 new cases of BC, 8195 deaths, and 102,223 prevalent cases (5-year) in 2022 [2].

BC is the leading cause of death from neoplastic diseases in women, with an increasing number of cases due to risk factors such as obesity and aging [3].

Obesity is linked to BC by associated mediators such as endocrine, inflammatory, and oxidative stress that modify the tissue microenvironment [4,5,6,7]. There is evidence of an increased risk of BC recurrence and mortality in both pre- and post-menopausal women with obesity compared with normal body mass index (BMI) women [8]. Also, age is linked to BC pathogenesis because cell aging increases the risk of developing cancer [9,10].

Estrogen exposure is one of the main contributors to BC, as it directly correlates with BC risk (early menarche and or delayed menopause has an increased risk). On the other hand, pregnancy and lactation, due to the decreased levels of estrogen in these periods, reduce BC risk [11].

BC is considered a complex disease. Environmental and genetic factors contribute to BC carcinogenesis. Among the genetic risk factors involved in sporadic cancer, single-nucleotide polymorphisms (SNPs) are the most common type of genomic variant (over 10 million), and some of them have been demonstrated to contribute to BC carcinogenesis [12].

Genome-wide association studies (GWAS) have identified over 1771 SNPs associated with a risk of BC (NHGRI-EBI GWAS catalog) on ClinVar, and over 1887 on 21 May 2024 in ENSEMBL [13]. Most GWAS studies are based on European populations, while the Latin-American population remains underrepresented [14]. The Latin-American population is characterized by a broad admixture between European, indigenous American, and African ancestry [15,16]. For this reason, there is a need for a better approach for this population, including the employment of multi-locus models (multidimensional) in GWAS analysis to recover a model closer to reality.

SNPs often explain normal variation between individuals and frequently have minimal functional impact. BC-associated SNPs individually contribute a low risk, but when considered together they lead to a cumulative risk [17].

Due to the multifactorial nature of BC, our goal was to explore the relationship between a set of SNPs previously reported as risk factors for BC, including metabolic features and informative ancestry markers in BC patients from Northeastern Mexico. This study incorporates BMI, menopause status, and age [5] as cofactors, analyzing gene–environment (G × E) interactions using a multi-locus statistical model.

## 2. Materials and Methods

### 2.1. Scientific and Ethics Committees Approval

The study was conducted following the Declaration of Helsinki, and the protocol was approved by the Institutional Ethics Committee of the University Hospital, with the registration codes BI10–002 and GN13-001. All subjects (patients and controls) were invited to participate in the research project. An interview was performed, and once the subjects agreed to participate, they signed an informed consent letter. Afterward, clinical and epidemiological information was collected, and blood samples were taken.

### 2.2. Study Design and Population

The study was retrospective and nested case-control. All the participants were Mexican women whose four generations were born in Northeast Mexico (Nuevo León, San Luis Potosi, Zacatecas, Coahuila, and Tamaulipas). Women with a family history of BC and pregnant women were excluded.

### 2.3. Patients Group

A total of 92 BC patients were selected among women receiving chemotherapy at two institutions in Monterrey City, Mexico: the Centro Universitario Contra el Cáncer (Hospital Universitario “Dr. José E. González” (HU), of the Universidad Autónoma de Nuevo León (UANL), and the Hospital de Especialidades #25 (Instituto Mexicano del Seguro Social (IMSS)). Both are Mexican Northeast referral centers.

We included only patients with sporadic BC. All BC cases were confirmed by histopathologic analysis.

### 2.4. Controls Group

Controls (*n* = 126) were women older than 44 years of age, without a personal history of cancer, and a BI-RADS 1–2 mammogram. Women with a family history of BC and who were pregnant were excluded.

### 2.5. Nucleic Acid Extraction

DNA was extracted from 200 µL EDTA-treated whole blood samples using the QIAamp^®^ Blood Mini Kit on the automated QIAcube system (Qiagen, Hilden, Germany) according to manufacturer instructions. Purified DNA was collected in a final volume of 150 µL and stored at −20 °C until use. Samples with low DNA quality extracted (1.7 ratios at 260/280 nm) were excluded from the study [18].

### 2.6. SNP Selection

We selected 121 SNPs categorized into four groups (Appendix A): Risk factors for BC (*n* = 22), SNPs related to glucose-associated metabolic pathways (*n* = 50), SNPs related to both BC and glucose-associated metabolic pathways (*n* = 13), and ancestry markers (*n* = 36). The selected SNPs were located in 48 genes and 6 intergenic regions. SNPs from 48 genes include: *ANKK1*, *CAPN10*, *CCL*, *CD36*, *CDKAL1*, *ENPP1*, *ESR1*, *FABP2*, *FAS*, *FTO*, *GRIK4*, *HHEX*, *HNF1B*, *HNF4A*, *hsa-miR-196a2*, *IGF2BP2*, *IL10*, *IL1B*, *IL6*, *IL8*, *INHBB*, *KCNQ1*, *LEPR*, *LGALS3*, *LTA*, *MC4R*, *MMP2*, *MMP7*, *MMP8*, *MTHFR*, *NEGR1*, *NEUROD1*, *PCK1*, *PON1*, *PPARG*, *RPTOR*, *SCIN*, *SFRS15*, *SLC30A8*, *TCF7L2*, *TNF*, *UCP1*, *UCP2*, *UCP3*, *WFS1*, *WNT5B*, *ZNF365*, and *ZNF703*. We also included six intergenic regions: *CDKN2B-AS1*: *DMRTA1*, *CDC123*: *CAMK1D*, *SCT*: *DRD4*, *KCNJ11*: *ABCC8*, *USF1*: *TSTD1*: *ARHGAP30*, and *VLDLR*: *FLJ35024*. Ancestry Informative Markers (AIMs) were included to determine the population structure [19,20].

### 2.7. Genotype Analysis

Molecular analysis was performed using TaqMan^®^ assays (Applied Biosystems, Carlsbad, CA, USA) and analyzed in an OpenArray^®^ NT Genotyping System (Applied Biosystems, Carlsbad, CA, USA) (Appendix A) [18]. We calculated the accuracy of the genotyping by comparison with concordance calls generated for 10 samples genotyped three times. Samples with low-quality genotypes (Sample call rate ≥90%; SNPs per sample/the total number of SNPs in the dataset) were excluded from the study (Appendix A) [18].

### 2.8. Statistical Methods

We calculated genotype statistics by marker and sample. Samples with inbreeding and a call rate <90% were excluded, as were markers with a call rate < 88%. Hardy–Weinberg Equilibrium (HWE) was analyzed for each SNP (*p*-values < 0.01 were excluded).

For the association study, a multi-locus mixed model (MLMM) analysis was performed using SNP and Variation Suite v8 (Golden Helix, Inc., Bozeman, MT, USA) [21,22,23]. We used demographic data and 10 genetic principal component analyses (PCA) as covariants [23] (Figure 1).

The main factors considered in the statistical design included the number of genetic markers, the sample size from cases and controls, and the overall study design. Our analytical model was specifically developed to minimize false positives and eliminate potential confounding among the studied markers by incorporating adjustments for population structure, kinship, and cryptic relatedness [22,23].

After obtaining the initial results from the multi-locus mixed-model (MLMM) analysis, we selected the optimal model steps using several correction methods: multiple Bonferroni correction, the posterior probability of association (PPA), as well as modified and extended Bayesian information criteria (BIC) [24]. We then evaluated the model’s performance using quantile–quantile (Q-Q) plots to ensure that our *p*-values were not inflated due to multiple tests. We considered the individual false discovery rate (FDR) and the Bonferroni-adjusted *p*-values for the final model selection, testing under additive, dominant, and recessive genetic models.

Additionally, we generated quantile–quantile (Q-Q) plots for additive, dominant, and recessive models via the SNP and Variation Suite v8 (Golden Helix, Inc., Bozeman, MT, USA) [22].

To elucidate the potential biological impact of the identified genetic variants, we performed a series of functional analyses using several bioinformatics tools.

### 2.9. In Silico Variant Effect Predictor Analysis

The Ensembl-VEP bioinformatics platform (Cambridge, UK, https://www.ensembl.org/Homo_sapiens/Info/Index, (accessed on 6 February 2024)) was used to classify regulatory region variants, nonsense-mediated decay transcript, and missense variants [25]. The analysis was based on the human genome annotation GRCh37/hg19.

### 2.10. Protein–RNA Interaction Prediction Using RNAct

For non-coding variants that may influence RNA regulation, we utilized the RNAct database (http://rnact.crg.eu (accessed on 12 February 2024)). This tool was employed to predict potential protein–RNA interactions, helping us infer possible regulatory mechanisms that could be affected by the variants [26].

### 2.11. Pathway and Gene Ontology Enrichment with ToppGene Suite

Finally, we applied the ToppGene Suite to perform enrichment analysis. We submitted the list of genes correlated with our identified variants to ToppGene Suite (Cincinnati, OH, USA, https://toppgene.cchmc.org/ (accessed on 12 February 2024)), which enabled us to identify overrepresented biological processes, molecular functions, and pathways. This analysis provided insights into how these genes—and, by extension, the variants—might contribute to BC risk through alterations in pathways related to metabolism, inflammation, and extracellular matrix remodeling.

## 3. Results

### 3.1. Patients Data

Our cohort consisted of 218 women, including 92 BC cases and 126 unaffected controls. The age range of patients and unaffected controls was between 45 and 72 years old. There were no statistically significant differences between the groups regarding age at menarche, use of oral contraceptives, and number of children (*p* > 0.05). Statistically significant differences were found for increased age (*p* = 0.023), reduced height (*p* = 0.048), elevated BMI (*p* = 0.011), younger age at menopause (*p* = 0.025), and confirmed menopausal status (*p* = 0.004) in BC cases compared to controls. The anthropometric and physiological data of the participants are detailed in Table 1. The immunohistochemistry analysis of the BC patients is detailed in Table 2.

### 3.2. Genotype Analysis Results

From the 128 SNPs considered, 121 SNPs were eligible for statistical analysis, and seven SNPs were excluded due to low quality. No significant genomic inflation (expresses the deviation of the distribution results observed compared to the distribution of the expected test statistic [24,27]) was detected between the observed and expected *p*-values in the MLMM (polygenic inheritance) under additive (Appendix A), dominant (Appendix A), and recessive (Appendix A) genetic models (Figure 2 and Figure 3, Table 2) [24]. The genotype frequencies and HWE were reported in 12 BC risk-associated SNPs, as these showed statistical significance (Table 3).

### 3.3. MLMM Testing

A significant association between BC and 12 gene variants was identified in the following genes: rs3856806 (*PPARG*), rs3917542 (*PON1*), rs3750804 and rs3750805 (*TCF7L2*), rs12792229 (*MMP8*), rs1121980 and rs3751812 (*FTO*), rs12946618 (*RPTOR*), rs2833483 (*SCAF4*), rs1800955 (*SCT-DEAF1-DRD4*), rs5218 (*KCNJ11*-*ABCC8*), and rs11652805 (*AMZ2P1*-*GNA13*) (Table 4).

Figure 2 illustrates the –log10 of FDR *p*-values under the Q-Q’s additive, dominant, and recessive models, considering the expected and the observed *p*-values. Figure 2A plots the cluster of samples based on the similarity of two main genetic principal components. There was no evidence of substructure in the populations of the cases and control groups. Figure 2A reveals a homogeneous distribution of cases (green dots) and controls (blue dots). Figure 2B shows Q-Q plots for additive genetic model results (step 117 selected by Multiple PPA and Multiple Bonferroni). Figure 2C shows the dominant genetic model results (step 117, chosen by Multiple and Bonferroni). Figure 2D shows the recessive genetic model results (step 115 selected by Modified Bonferroni and Extended Bonferroni). *TCF7L2* and *FTO* variants were associated in the additive genetic model, *RPTOR* and *MMP8* in the dominant genetic model, and *KCNJ11*, *RPTOR*, *PON1*, and *PPARG* variants in the recessive genetic model.

Figure 3 shows the results of the Manhattan plots by relative chromosome position of the association study in additive, dominant, and recessive models with MLMM.

Of the 12 highly BC-associated SNPs, 10 are located in non-coding regions (Table 5) involved in the decay of mRNA transcripts, enhancers, promoters, and regulatory regions (nonsense-mediated decay transcript variants (NMDTV), promoter flanking region (PFR), and regulatory region variants (RRV)).

Variants in the *MMP8* (rs12792229) and the *PPARG* (rs3856806) genes have a dual effect (refers to the fact that these variants are located in regions that can influence both the protein-coding sequence and regulatory elements of the gene) in coding and non-coding regions about the alternative transcripts of the genes [28] (Table 6).

For further understanding about the pathologic consequences of SNPs reported in this work, Table 7 provides a summary of the significant SNP findings across various population studies, and Table 8 describes the gene function.

In our functional analysis inference, we first employed Ensembl’s Variant Effect Predictor (VEP) to annotate each genetic variant, classifying them into categories such as missense, synonymous, and regulatory region variants (Appendix A). For non-coding variants with potential regulatory roles, we utilized the RNAct database to predict protein–RNA interactions, thereby providing insight into how these variants might influence RNA stability and gene expression. Finally, we conducted pathway and gene ontology enrichment analysis using the ToppGene Suite. This analysis revealed a significant overrepresentation of biological processes and molecular functions related to adipogenesis, inflammation, and extracellular matrix remodeling. Together, these results suggest that the identified genetic variants may modulate BC risk by affecting key metabolic and inflammatory pathways, offering a comprehensive view of their functional relevance in the context of BC development. Table 9 shows the functional analysis inference checklist results. Table 10 shows gene correlations, expression patterns, and functional implications for BC-associated SNPs based on Ensembl, RNAct, and ToppGene results.

## 4. Discussion

The current study investigated the association between BC and epidemiological data through a case and control study. The age difference between the two groups is statistically significant; it can be explained by the recruitment process for the control group, which consisted of women undergoing mammography screening. Those with negative results were invited to participate as controls. Our age findings are consistent with a previous study conducted at the National Cancer Institute (Mexico), which reported that the average age of women (N = 10,000) undergoing mammograms was 52.61 years old, ranging from 39 to 78 years old. In that study, the average age of patients with a positive diagnosis was 56 years old, compared to 51.6 years old for those with negative results (*p* = 0.023) [37].

Although there were no significant differences in age at menarche, oral contraceptive use, and children number in the two groups, several factors emerged as potential contributors to BC susceptibility. One of the most notable findings was BMI distribution. While the cohort as a whole presented a predominance of overweight status, this trend was more pronounced in the BC group, revealing a statistically significant association between increased BMI and BC (*p* = 0.011). Our results are consistent with the consensus suggesting that elevated BMI is associated with increased estrogen levels due to adipose tissue functioning as an additional source of estrogen production [38]. This link becomes especially pertinent in post-menopausal women, where excess adipose tissue can lead to prolonged estrogen exposure, thus contributing to cancer development.

Mexican demographics directly impact the BC risk by age and BMI. Population projections for the coming decades indicate an increase in individuals aged 50–60 years old [39]. Additionally, Mexico is considered one of the countries with the highest rates of overweight status and obesity (World Obesity Federation, World Obesity Atlas 2023. https://data.worldobesity.org/publications/?cat=19 (accessed on 13 June 2024)), especially among women, with over 70% being affected by overweight status or obesity [40]. These facts reflect a critical health issue that potentially impacts BC incidence. Other risk factors not influenced by ethnicity include menarche, menopause status, age of first pregnancy, and number of pregnancies [5].

In the Northeast region of Mexico (Nuevo León, San Luis Potosi, Zacatecas, Tamaulipas, and Coahuila), the distribution of BC subtypes generally aligns with national trends; however, regional variations may still be worth considering. Based on available studies and data [41,42,43,44], the approximate distribution of BC subtypes in this region is as follows: Triple-Negative BC (TNBC) ~10–20% frequency (present study 19.6%); this subtype is prevalent in the Northeast and other regions of Mexico, and is especially significant among younger women. HER2-positive subtype ~10–15%, the prevalence of this subtype is also notable in the Northeast region (present study 7.6%). This distribution of BC subtypes is a general overview, but the precise distribution may vary slightly depending on specific studies and population groups within the region. Efforts for local cancer registries, hospital data, and regional epidemiological studies would offer the most accurate and detailed information for the Northeast of Mexico.

Previous genome-wide association studies (GWAS) have identified SNPs associated with the risk of BC and obesity [13,45]. Our study analyzed 121 SNPs and found 12 highly associated with an increased risk of developing BC. Statistical analysis for SNP classification was designed to minimize false positives and eliminate confusion factors, considering population, kinship, and cryptic relation.

One of the common problems in genome-wide studies is that the population structure needs to be considered, especially in ethnically diverse and complex populations [46,47]. Including AIMs in this study ensures that a mestizo population is well-represented and that population bias is excluded. The present study included 36 AIMs (see Appendix A for further details), and there was no evidence of substructure in the populations of the cases and control groups.

Several projects and articles were reported in GWAS studies of BC for women of European and African ancestry [48,49,50]. However, a low proportion of projects and articles reported BC GWAS in other populations.

We found an association of 12 SNPs with BC; two are located in coding and 10 in non-coding regions. These results align with expectations, as most SNPs and CNAs (Copy Number Alterations) are located in non-coding sequences, where they regulate oncogenes and tumor-suppressor genes in the carcinogenesis process [51]. Products of the associated genes in this work involve a variety of functions such as transcriptional factors and regulators, lipid metabolism, remodeling and modifying DNA, cell growth, RNA polymerase modification, ionic channels, and addiction-related receptors. The effects of these SNPs are listed in Table 5 and Table 6.

To address the associated variants in the present study, we individually discuss each variant and corresponding gene and their implications for BC.

### 4.1. The rs11652805 (AMZ2P1; GNA13 Intergenic Variant, Regulatory Region Variant)

The intergenic variant located between *AMZ2P1* and the *GNA13* and contiguous to a regulatory region (enhancer) is part of the sequence of a transcription factor binding site (ENSEMBL database) [13]. This regulatory variant could be involved in the transcriptional regulation of *AMZ2P1* and *GNA13*; evidence supports the gene expression correlations between rs11652805 and *AMZ2P1* (ENSG00000214174) and *GNA13* (ENSG00000120063) transcript expression [13].

#### 4.1.1. *AMZ2P1* (Archaelysin Family Metallopeptidase 2 Pseudogene 1)

The AMZ2P1 (Archaelysin Family Metallopeptidase 2 Pseudogene 1) is classified as a pseudogene. However, ENSEMBL reported an unprocessed mRNA transcript, and 19 lncRNA biotypes could be functionally relevant [13]. The lncRNA ENST00000397713.5 related to rs11652805 of *AMZ2P1* was analyzed for protein–RNA interaction predictions in the RNAct database (http://rnact.crg.eu (accessed on 12 February 2024)) [26], as a result (in silico) predicted interactions by protein–RNA interaction prediction algorithm with 100 protein-coded genes. The primary function classification (ToppGene, https://toppgene.cchmc.org/ (accessed on 12 February 2024)) of the 100 genes that interact is molecular function: ATP hydrolysis activity (17/100), ATP-dependent activity (20/100), and biological process; cell cycle process (24/100), chromatin remodeling (16/100) [52].

*GNA13* (Guanine nucleotide binding protein alpha 13) is predicted to enable D5 dopamine receptor binding activity, G-protein beta/gamma-subunit complex binding activity, and GTPase activity [36]. *GNA13* promotes tumor cell invasion and metastasis by activating the RhoA/ROCK signaling pathway [53,54,55] and the GPCR, BC regulation by Stathmin1, and Estrogen Pathway [36]. *GNA13* expression in BC cells is regulated by post-transcriptional mechanisms involving miR-31 [56]. *MiR-31* partially controls BC cell invasion by targeting *GNA13*. The loss of *miR-31* and increased *GNA13* expression may serve as biomarkers for BC progression [56]. Increased *GNA13* expression has been observed in metastatic BC cells. In BC, enhanced *GNA13* signaling represses *KLK* gene expression [57]. We found the association of the rs11652805 and BC; this SNP has no previous record of association with cancer risk. However, the proximity localization to a regulatory region may be implicated in regulating gene expression of contiguous genes, probably *AMZ2P1* and *GNA13*, as context; the chromatin stretch enhancer states can drive the cell-specific gene regulation in human disease risk variants [58,59].

The *GNA13* RNA transcript ENST00000439174.6 was analyzed in the RNAct database [26], as a result (in silico) predicted interactions by protein–RNA interaction prediction algorithm with 100 protein-coded genes. The primary function classification (ToppGene) of the 100 genes that interact is molecular function: chromatin binding (16/100), histone binding (8/100), and Biological Process; chromatin organization (20/100), protein-DNA complex organization (20/100), and chromatin remodeling (16/100) [52].

The *GNA13* and *AMZ2P1* RNA transcripts interact with protein-coded genes related to chromatin remodeling.

#### 4.1.2. DRD4 (Dopamine Receptor D4)

The gene encodes a D4 subtype dopamine receptor coupled to G-protein. *DRD4* is responsible for neuronal signaling in the brain’s mesolimbic system, inhibiting adenylyl cyclase [36]. In one study, *DRD4* was overexpressed in BC cells compared to normal tissue. In the same study, MCF-7 and MDA-MB-231 cells, only *DRD1* and *DRD4* proteins were detected [60]. With decades of evidence suggesting a link between dopamine (DA) receptors and cancer, the DA pathway has recently emerged as a potential target in antitumor therapies [61]. The dopamine receptors (DAR) are associated with tumor cell death, proliferation, invasion, and migration. The DAR could regulate several ways of tumor cell death (apoptosis, autophagy-induced death, and ferroptosis) [62]. We found an association between rs1800955 and BC risk not previously reported; based on the relation of *DRD4* with cancer biology, this DMA variant rs1800955 association is probably not casual and maybe could be helpful to identify BC patients and pharmacology strategies recommendations in further studies. Another possible impact in BC is that the rs1800955 SNP is an intergenic flanking promotor variant in a transcription factor binding site at the 5’ of the *DRD4* gene [13]. Genes regulating *DRD4* transcript expression are involved in the cell cycle process (25/100) and chromatin remodeling (14/100) [52].

### 4.2. FTO (Fat Mass and Obesity-Associated Protein)

*FTO* gene is part of the AlkB-related non-heme iron and 2-oxoglutarate-dependent oxygenase superfamily. Its gene product has RNA demethylase activity and regulates adipose and energy homeostasis [36]. FTO is related to obesity and diabetes predisposition [63]. Some studies have related *FTO* SNPs with various types of cancer [34], including BC, where the overexpression of *FTO* promotes cell glycolysis and impacts the PI3K/Akt signaling pathway [64]. There has been multiple SNP related to cancer (rs9939609, rs8050136, rs1477196, rs6499640, rs1121980, rs17817449, rs11075995, rs8047395, and rs7206790, rs3751812) [34]. The most studied is rs9939609, associated with lung cancer, renal cancer, BC, prostate cancer, pancreatic cancer, and endometrial cancer [65,66,67,68,69]. In a Chinese cohort of women with BMI < 24 kg/m2, rs1477196 AA genotype and TAC haplotype for rs9939609, rs1477196, and rs1121980 were found to have a protective effect for BC, and SNP rs16953002 AA genotype was found to have an increased BC risk [70]. A meta-analysis found an association between rs11075995 and ER-negative BC in the European population and rs17817449 with a generally increased risk for BC in the African population [71]. In our study, we found rs3751812 and rs1121980 associated with BC as seen in other populations, and there was no association for the rs10938397, rs17817449, rs2815752, and rs8050136 in our population regardless of the BMI. The rs1121980 and rs3751812 were classified as regulatory region variants, RRV, each variant was located in a different regulatory feature, the rs1121980 located in the regulatory feature ENSR00001001829, and the rs3751812 in the regulatory feature ENSR00000537825 (Ensembl database) [13]. The DNA variants linked to two regulatory regions were gene expression correlations (Ensembl database) to genes functionally implicated in the mRNA N6-methyladenosine dioxygenase activity (ToppGene, https://toppgene.cchmc.org/ (accessed on 12 February 2024)) [52], which plays an important role in mediating fundamental pathological and physiological metabolic processes, processing, translation, and stability of RNA [72], thromboxane A2 receptor binding, thromboxane A2 (*TXA2*) is a potent lipid mediator released by platelets and inflammatory cells and is capable of inducing vasoconstriction [73], tRNA demethylase activity. Additionally, the rs1121980 is a nonsense-mediated decay transcript variant that NMD surveys newly synthesized mRNAs and degrades those that harbor a premature termination codon (PTC), thereby preventing the production of truncated proteins that could result in disease in humans [74].

### 4.3. KCNJ11 (Potassium Voltage-Gated Channel Subfamily J Member)

This gene is a member of the potassium channel gene family, controlled by G proteins. *KCNJ11* is associated with diabetes and cardiac conduction defects [36]. *KCNJ11* gene has more than 200 polymorphisms; six of them, rs5215, rs5210, rs5218, rs5219, rs886288, and rs2285676, are related to diabetes [75]. The rs5219 has been previously associated with colon cancer risk [76,77]. The high *KCNJ11* gene expression is associated with favorable prognostic and life expectancy in renal cancer [78] (data available from The Human Protein Atlas, version 23.0). Our work found an association of rs5218 (A190A) and BC. This SNP has been associated with Type II diabetes mellitus [79], but no previous evidence exists with BC risk. The rs5218 is a synonymous variant for *KCNJ11* and the regulatory region variant, a CTCF binding site variant for *KCNJ11*, *ABCC8*, and *NCR3LG1* (ENSR00000263063) [13]. NCR3LG1 is tumor overexpressed and is associated with fatal disease progression of various cancers [80]. *ABCC8* (SUR1) sulfonylurea receptor 1 is a tumor-enhancer in non-small cell lung carcinoma (NSCLC) [81].

The rs5218 SNP is a regulatory region variant in a transcription factor binding site (regulatory feature ENSR00000263063) [13]. The rs1800955 has a gene expression correlation (*p*-value < 0.001) in the ENSEMBL database (https://www.ensembl.org) (accessed on 4 February 2024) with 25 genes in different tissues [13]. The analysis of the 25 genes expressed in ToppGene results in a molecular function identity. The study of the 25 genes expressed in ToppGene results in a molecular function identity in a voltage-gated potassium channel activity (*KCNC1*, *ABCC8*, and *KCNJ11*) and a WikiPathways VITAMIN B12 METABOLISM, and FOLATE METABOLISM [52].

### 4.4. MMP8 (Matrix Metalloproteinase 8)

MMPs are members of a large multigene family of zinc-dependent endopeptidases involved in remodeling the extracellular matrix protein degradation of proteins to allow ductal progression across the basement membrane [82,83,84]. *MMP8* is also known as collagenase-2 or neutrophil collagenase. Tumorigenic and antitumorigenic properties have been attributed to MMP8 activity in cases of head and neck squamous carcinoma cells [85]. In BC, *MMP8* gene variation may influence prognosis and could have an inhibitory effect on cancer metastasis; the minor allele (T) of the promoter SNP (rs11225395) has been linked to a better prognosis, such as reduced susceptibility to lymph node metastasis [86], reduced cancer relapse [87], and higher survival [87]. In this work, we found the rs12792229 of *MMP8* associated with BC. There are no reports of clinical significance about this SNP; further studies are required to understand its role in BC development.

### 4.5. PON1 (Human Serum Paraoxonase 1 Enzyme)

This gene is a member of the paraoxonase family of enzymes with lactonase and ester hydrolase activity. PON1 is involved in HDL regulation [36,88,89]. *PON1* SNPs have been related to multiple inflammatory diseases, such as atherosclerosis, diabetes, and some cancer types, including lung, multiple myeloma, papillary thyroid cancer, and prostate, breast, and ovarian cancer [90]. *PON1* rs854555 was also related to an increased risk of BC in U.S. post-menopausal women [32]. Another study found that rs854560, rs662, rs705379, and PON1_304A/G were associated with an increased risk of developing cancer in Caucasian and Asian populations [30]. We included rs3917542, rs854555, and rs662, but only the rs3917542 was associated with BC. *PON1* rs3917542 has been associated with cardiovascular diseases but not cancer [91]. *PON1* polymorphism association studies in cancer differ among populations; there are no studies on the Mexican population [90]. Increased expression of *PON1* was correlated with higher survival probability in liver cancer [78] (data available from The Human Protein Atlas, version 23.0).

The rs3917542 has a gene expression correlation (*p*-value < 0.05) in the ENSEMBL database (https://www.ensembl.org) (accessed on 4 February 2024) with 14 genes in different tissues, breast mammary tissue included [13]. The analysis of the 14 genes expressed in ToppGene results in a molecular function identity. The ToppGene results in a molecular function identity in arylesterase and lactonohydrolase activity [52].

### 4.6. PPARG (Peroxisome Proliferator-Activated Receptor Gamma)

The *PPARG* is a nuclear hormone receptor of the family of transcriptional regulators activated by ligands and involved in regulating adipogenesis and glucose [92]. The *PPARG* high expression in renal and urothelial cancer was associated with favorable life expectancy and unfavorable life expectancy in liver cancer (https://www.proteinatlas.org/ENSG00000132170-PPARG/pathology, accessed on 12 February 2024) *PPARG* SNPs have been associated with cancer risk, in particular for BC, ovarian carcinoma, follicular lymphoma, and colorectal cancer [92,93,94]. We analyzed rs3856806 and rs1801282 for this gene and found that rs3856806 was associated with BC development. Our results are concordant with another study in the Turkish population, where they found that both SNPs were in linkage disequilibrium and related to increased BC risk [95].

Conversely, a meta-analysis study found that rs3856806 was not associated with BC, and rs1801282 has a protective factor in Caucasian women [31]. PPARG expression as a prognostic effect depends on metastasis localization in advanced colorectal cancer patients. Increased expression of *PPARG* high expression increased the malignancy-associated traits such as proliferation in colorectal cancer cell lines and increased sensitivity toward the chemotherapeutic agent 5-FU [96]. The combination therapy of *PPARG* agonists and 5-FU-based chemotherapy is a promising strategy [96].

The rs3856806 has a gene expression correlation (*p*-value < 0.05) in the ENSEMBL database (https://www.ensembl.org) (accessed on 4 February 2024) with 15 genes in different tissues, breast mammary tissue included [13]. The analysis of the 15 genes expressed in ToppGene did not find statistically significant results [52].

### 4.7. RPTOR (Regulatory Associated Protein of MTOR Complex 1, RPTOR, Also Named RAPTOR)

*RPTOR* encodes a protein part of the PI3K–AKT–mTOR pathway, regulating negatively the mTOR kinase [36]. The PI3K pathway is frequently dysregulated in cancer, with multiple targeted therapies available and more in development [97]. The mammalian target of rapamycin complex 1 (mTORC1) is evolutionally conserved and commonly activated in various tumors [98]. Akt/Raptor signaling upregulation is associated with rapamycin resistance of BC cells [99]. Hypomethylation of CpG sites in *RPTOR* in blood had been associated with BC, suggesting that it might serve as a biomarker [100]. *RAPTOR* gene polymorphisms (rs11653499 and rs7212142) were significantly related to the risk of urothelial cancer [101]. Cheng, T.Y et al. report that rs9900506 and rs3817293 have been linked to a protective effect for BC in European-American and African-American women, respectively [102]. We found rs12946618 associated with BC. To our knowledge, no previous reports of this SNP and BC risk exist. The rs12946618 is a regulatory region variant (regulatory feature ENSR00001345149) correlated with gene expression of 33 genes in different tissues (*p*-value < 0.05) in the ENSEMBL database [13]. The ToppGene analysis of the 33 genes expressed correlated results in a molecular function in a single-stranded RNA binding, methylated histone binding, and cellular component PRC1 complex, a protein complex that includes a ubiquitin-protein ligase and enables ubiquitin-protein ligase activity (EMBL-EBI database, GO:0000151), the PcG protein complex; transcriptional repressors that regulate several crucial developmental and physiological processes [103]. The related pathways are Reactome regulation of *PTEN* gene transcription and Reactome sumoylation of DNA methylation proteins. In the present study, we found LD with the rs12946115 and rs12950541, located in the *RPTOR* gene. VEP identified the rs12946618, rs12946115, and rs12950541 as Nonsense-mediated decay transcript variants (NMDTV) in the ENST00000574767.5 (name RPTOR-208) transcript, which codifies to a peptide of 95 amino acids; the canonical protein has 1335 amino acids (Transcript ENST00000306801.8; name RPTOR-201). The *RPTOR* rs12946618, rs12946115, and rs12950541 were classified as NMDTV; these variants can regulate *RPTOR* gene expression. *RPTOR* rs12950541 is an SNV (single-nucleotide variant) intronic variant that impacts alternative splicing, interfering with splice site recognition, affecting the synthesis of the protein, and thus interfering with its function. RPTOR is involved in nutrient signaling, mitochondrial oxygen consumption, and oxidative capacity [104].

RPTOR is involved in insulin receptor signaling and the AMPK, mTORC1, and metformin pathways. Complex 1 of mTOR (mTORC1) controls cell proliferation, growth, and metabolism in various cell types through a complex signaling network. To compensate for hyperglycemia, pancreatic β cells become hyperplasic hypertrophic, insulin secretion is increased [105], and finally, β cells begin to fail, and hypoinsulinemia appears. During T2D progression, complex 1 of the mammalian target of rapamycin (mTORC1) is hyperactivated, with a benefit due to an increase in β cell proliferation. Nevertheless, the chronic mTORC1 hyperactivation increases endoplasmic reticulum (ER) and Golgi apparatus stress and β cell failure [105,106].

With the in silico analysis of the *RPTOR* the rs12946618, rs12946115, and rs12950541 variants, we found that the degradation of mRNA *RPTOR* via the NMD process, and as a consequence, a deficient expression level of RPTOR protein in homozygous and heterozygous carriers, can be the factor associated with T2D development because RPTOR is required for maintaining a β-cell identity as well as repressing β-cell to α-cell transdifferentiation [107]. Furthermore, in a high caloric diet, mainly a high carbohydrate diet, mTORC1 is hyperactivated, and the rs12950541 variant can play a role in insulin resistance, obesity, and T2D development by loss of maintaining β-cell identity [107].

### 4.8. KCNJ11 Pathway and RPTOR

Raptor/mTORC1 is essential for b-cell function, and mTORC1 and mTORC2 regulate insulin secretion through Akt in INS-1 cells [108].

### 4.9. SCAF4 (SR-Related C-Terminal Domain-Associated Factor 4)

*SCAF4* encodes a member of the arginine/serine-rich splicing factor family. It is predicted to bind to the serine-phosphorylated c-terminal domain of POLR2A during RNA transcription and to be involved in RNA splicing [36]. Increased expression of *SCAF4* is related to a poor survival probability in BC (https://www.proteinatlas.org/ENSG00000156304-SCAF4/pathology/breast+cancer) (accessed on 12 February 2024). In another study, a subtype of signature composed of ELOA and SCAF4 was identified, and a subtype diagnostic and prognostic model was constructed for therapeutic targeting in esophageal cancer [109]. We found rs2833483 associated with BC. Based on recent studies, SCAF4 is a target and biomarker of interest in cancer. The rs2833483 has a gene expression correlation (*p*-value < 0.05) in the ENSEMBL database (https://www.ensembl.org) (accessed on 12 February 2024) with 17 genes in different tissues [13], and a negative gene expression correlation with *ENSG00000273091* (lincRNA gene), the *ENSG00000273091* has a interaction with AEBP2 protein in breast mammary tissue [26], AEBP2 protein acts as an accessory subunit for the core PRC2 (Polycomb repressive complex 2), which mediates histone H3K27 (H3K27me3) trimethylation on chromatin leading to transcriptional repression [36].

### 4.10. TCF7L2 (Transcription Factor 7 Like 2)

The *TCF7L2* gene is involved in the Wnt signaling pathway. TCF7L2 is implicated in angiogenesis, cell proliferation, apoptosis, transcription regulation, and glucose homeostasis [13,36]. TCF7L2 has been associated with type 2 diabetes (T2D), chronic renal disease [110], and cancer, including BC and hepatocarcinoma [36,111,112]. In 2012, Connor et al. found in a Hispanic cohort, including Mexican women with BC, that rs7903146, rs3750805, rs7900150, and rs1225404 were associated with a higher risk of BC regarding ancestry [33]. Another study in the Egyptian population did not find an association between BC and rs12255372 [113]. Also, the TT genotype of the rs3750804 increased the BC risk in women with a history of diabetes but did not demonstrate significant interactions by ethnicity or genetic admixture [33]. We included in our study the *TCF7L2* rs10885390, rs11196175, rs7903146, rs10885406, rs7900150, rs12255372, rs3750804, rs3750805, rs290487 and rs1225404, and we found that the rs3750804 and rs3750805 were associated to BC for ages 45 to 79. The DNA variants rs3750804 and rs3750805 are classified as promoter flanking region (PFR) and regulatory region variants (RRV). rs3750804 and rs3750805 could be an impact in *TCF7L2* expression.

GWAS studies should include population structure analysis with geographical data in mestizo populations. Mexican native populations are ethnically and genetically diverse [47]. The DNA variants reported in this study could be validated in further studies and can be useful as biomarkers for response to chemotherapeutic treatment in BC patients. Mexico is a developing country where genetic tests are not routine, and SNP detection could help to support the potential benefits of personalized medicine for the Mexican population. Our results reinforce that individual studies by populations should be done as there are ethnic differences in the expression and effect of these SNPs.

However, the association between SNP alleles and phenotypic impact can be confounding due to linkage disequilibrium, which segregates driver DNA variants with passenger DNA variants (passively inherited but regulatory neutral) within populations. Because of this, it is necessary to analyze the SNPs and genes implicated and screen other gene variants that could participate in the disease’s manifestation. Consequently, GWAS should be validated by extensive and meticulous analysis to determine which of the associated SNPs are the actual driver factors of the disease [51].

### 4.11. GNA13 and RAPTOR, TCF7L2, SCAF4, KCNJ11, and FTO

With data from 60 invasive breast carcinoma cell lines of the DepMap portal (Expression Public 24Q2), a linear regression was obtained for *RPTOR* and *GNA13* (*p*-value 7.81 × 10^−5^) [114]. Similar statistically significant linear regression results were obtained for *GNA13* compared to *TCF7L2* (*p*-value 7.98 × 10^−4^), *SCAF4* (*p*-value 1.22 × 10^−3^), *KCNJ11* (*p*-value 1.22 × 10^−3^), *FTO* (*p*-value 2.27 × 10^−5^): However, no significant linear regression with the other genes was reported in this study.

### 4.12. GNA13 (Gα13) and RPTOR (mTORC1) Pathway Interaction

Germinal center b-cell-like diffuse large b-cell lymphoma (GCB-DLBCL) is characterized by the downregulation of *PTEN* and the activation of the PI3K signaling pathway. When *PTEN* is lost or repressed, phosphatidylinositol-3-phosphate accumulates, activating AKT and mTORC1, which promotes cell survival, proliferation, and growth. GCB-DLBCL is also associated with the loss of S1PR2 (sphingosine-1-phosphate receptor-2) and Gα13 (G-protein alpha 13, *GNA13* gene) signaling, negatively modulating GC b-cell migration and PI3K signaling. The constant activation of the PI3K/AKT/mTORC1 pathway in GCB-DLBCL is due to the loss or inactivation of PTEN and S1PR2-Gα13 signaling, which usually suppress cell survival, proliferation, and growth [115].

## 5. Limitations of This Study

It is necessary to state the limitations of our study. The main constraints encountered during the project were the following.

### 5.1. Sample Size and Study Design

The retrospective, case-control design and the relatively modest sample size (92 cases, 126 controls) limit the statistical power of our findings and may reduce the ability to detect smaller effect sizes. However, the population structure is homogeneous. The simultaneous analysis of 121 SNPs in both cases and controls provides valuable insights into the genetic composition of the studied population. Additionally, we could not assess causality or disease progression.

### 5.2. Geographical and Ethnic Specificity

All participants were recruited from a specific region in Northeastern Mexico, which may limit the generalizability of our conclusions to other populations. Although we employed AIMs to address population stratification, our findings may reflect unique genetic and environmental factors specific to this region.

### 5.3. Restricted Genetic Scope

We focused on 121 SNPs previously implicated in BC, metabolic disorders, and ancestry. While this targeted approach enabled an in-depth analysis of specific variants, it did not reflect the entire spectrum of possible genetic risk factors.

### 5.4. Limited Lifestyle and Environmental Data

Although we considered age, menopausal status, and BMI as cofactors, detailed information on diet, physical activity, alcohol consumption, and smoking was not systematically available for all participants. Leaked information limits our ability to assess fully the impact of gene–environment interactions on BC risk.

### 5.5. BC Treatment and Subtype Representation

Our study focused on 92 BC patients undergoing chemotherapy, which may inadvertently underrepresent patients with luminal A-like BC—typically managed with endocrine therapy alone. We acknowledge that this limitation could affect our findings, particularly about hormonal factors. Including patients receiving endocrine therapy in future studies would offer a more comprehensive and detailed perspective on BC subtypes in Northeastern Mexico while also increasing the sample size and subtype diversity.

### 5.6. Functional Validation

Our study centered on statistical associations; thus, functional assays to confirm or elucidate the biological impact of identified risk variants were beyond this project’s scope. Further molecular and mechanistic investigations are needed to validate the direct influence of these SNPs on BC development.

## 6. Conclusions

Our study identified associations between BC and specific gene variants (previously implicated in diabetes mellitus, obesity, insulin resistance, inflammation, and extracellular matrix remodeling), as well as clinical factors such as elevated BMI, younger age at menopause, and confirmed menopausal status, in a sample of Mexican women.

We found the association of 12 genetic variants located in the *DRD4*, *PPARG*, *PON1*, *TCF7L2*, *MMP8*, *FTO*, *RPTOR*, and *SCAF4* genes, and the non-coding regions in *SCT*, *DEAF1*, *KCNJ11*, *ABCC8*, *AMZ2P1*, and *GNA13*, with risk of BC, menopause, and BMI in Northeastern Mexico. Three variants are located in the exome, and ten are classified as non-coding, involved in the accelerated decay of the transcript, enhancer, promoter, and regulatory regions (Nonsense-mediated decay transcript variants), promoter flanking region, and regulatory region variants. Non-coding SNPs are found in regulatory regions, highlighting their potential impact on gene expression and regulation, potentially affecting pathways involved in cell cycle regulation, transcriptional activity, and oncogenic signaling, which may contribute to carcinogenesis.

The inclusion of AIMs in our study allows us to confirm that no population stratification bias was present, ensuring a well-represented mestizo population and enabling a more precise identification of SNPs relevant to the Mexican population. This approach could contribute to future decisions regarding personalized medicine tailored to the genetic profile.

Given the small sample size and potential selection bias, and considering that these SNPs are also linked to other endocrine and inflammatory diseases, our findings should be considered preliminary. These results suggest that gene–environment interactions may affect BC development in specific contexts. However, further studies in larger and more diverse cohorts are needed to validate these findings and better understand the functional impact of the identified SNPs on BC development and progression.

## Figures and Tables

**Figure 1 cancers-17-00982-f001:**
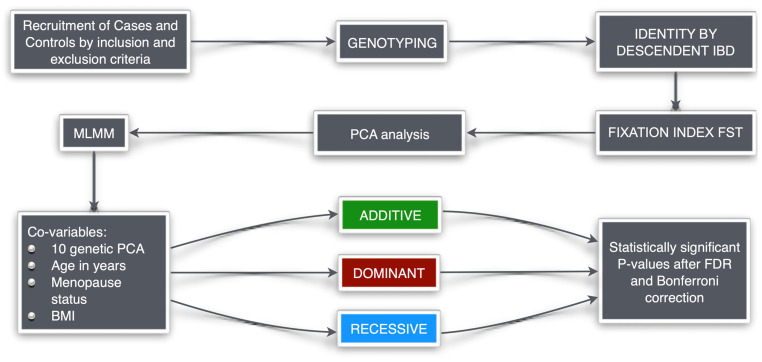
Schematic diagram of statistical design. Green box represents the additive genetic model, red box represents the dominant genetic model, and blue box represents the recessive genetic model.

**Figure 2 cancers-17-00982-f002:**
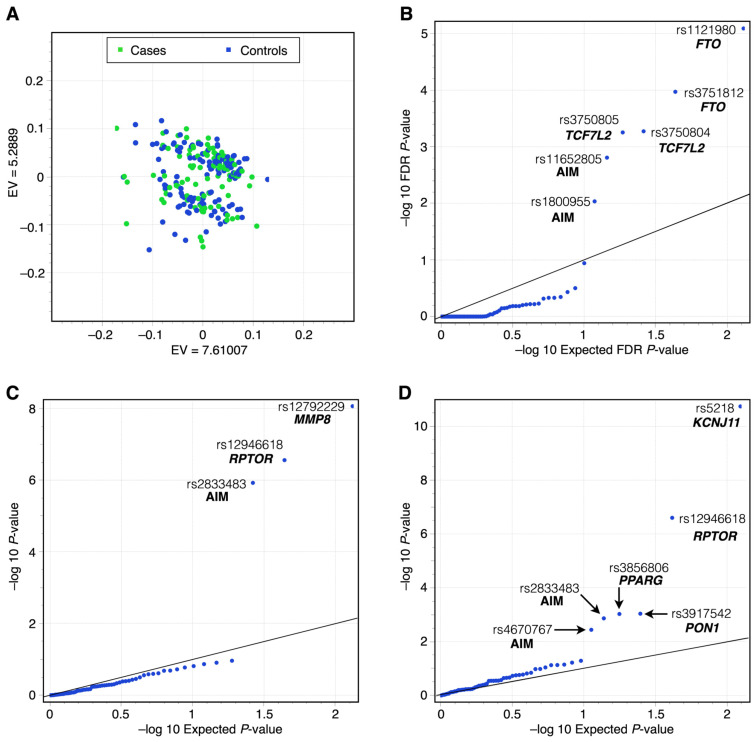
(**A**) Principal component analysis plot, expected vs. observed *p*-values in a P-P plot. There was no evidence of substructure in the populations of the cases and control groups, revealing a homogeneous distribution of cases (green dots) and controls (blue dots) (**B**) Additive genetic model step 117 selected by Multiple PPA and Multiple Bonferroni. (**C**) The dominant genetic model was chosen by Multiple and Bonferroni step 117. (**D**) Recessive genetic model step 115 selected by Modified Bonferroni and Extended Bonferroni. (Notes: (1) SNPs with significant *p*-values are located above the line of the respective genetic model (this line typically shows where SNPs with no association to the trait would be expected to fall. When SNPs fall above the line, it indicates that their observed *p*-values are lower than expected by chance, suggesting a significant association with the condition or trait being studied. (2) The selection of models helps determine how genetic variants contribute to BC risk. According to the results, each SNP agrees with one of the three models: additive (each additional risk allele has a proportional effect on the BC risk), dominant (having at least one copy of the risk allele is enough to influence the BC risk), and recessive (two copies of the risk allele are required to impact the trait or disease risk). Step refers to the iteration in the statistical analysis, which indicates the point at which the model was selected. Various correction methods (PPA, Bonferroni, Modified Bonferroni, Extended Bonferroni) were applied to control for false positives due to multiple tests).

**Figure 3 cancers-17-00982-f003:**
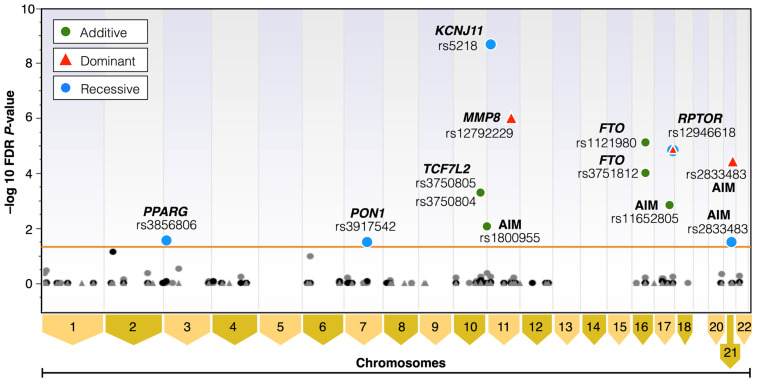
Manhattan plot results of the association study with MLMM in cases and controls age 45–79 years-log of FDR adjusted *p*-values. Manhattan plot helps to visually identify regions of the genome that are significantly associated with a particular trait or disease. X-axis represents the chromosomes, and Y-axis represents the negative logarithm of the *p*-value for each SNP. The horizontal orange line marks the significant *p*-value cut-off. Significantly associated SNPs marked as green dots represent the additive genetic model, red triangles represent the dominant genetic model, and blue dots represent the recessive genetic model. Non-significantly associated SNPs are marked as black dots in the additive genetic model, gray dots in the recessive genetic model, and gray triangles in the dominant genetic model.

**Table 1 cancers-17-00982-t001:** Anthropometric and physiological characteristics of cases and controls.

Characteristics	Controls (*n* = 126)	Cases (*n* = 92)	*p*-Value ^2^
Age (years) ^1^	54.93 ± 7.091	58.45 ± 8.811	0.023
Height (meters) ^1^	1.578 ± 0.068	1.558 ± 0.080	0.048
BMI ^1^	28.03 ± 4.950	29.92 ± 5.713	0.011
Menarche (Age, years) ^1^	12.63 ± 1.543	12.88 ± 1.503	0.231
Menopause (Age, years) ^1^	47.37 ± 5.630	45.29 ± 5.977	0.025
Menopause confirmed ^3^	83 (65.87%)	76 (82.61%)	0.004
Oral contraceptives ^3^	20 (15.87%)	23 (25.00%)	0.110
Children (number) ^3^	111 (88.10%)	82 (89.13%)	0.187

^1^ Results are presented as the mean ± standard deviation/standard error. ^2^ Unpaired t-test with Welch’s correction *p*-value. ^3^ Number and percentage values.

**Table 2 cancers-17-00982-t002:** BC tumor immunohistochemistry (*n* = 92), age range 45–79 years old.

Subtype	*n*	Percentage (%)	Proportion	95% CI (Proportion)	95% CI (Percentage)
TNBC (ER–/PR–/HER2–)	18	19.6%	0.1957	[0.1145, 0.2769]	[11.45%, 27.69%]
Strict HER2-Enriched (ER–/PR–/HER2+)) ^1^	7	7.6%	0.0761	[0.0220, 0.1302]	[2.20%, 13.02%]
Triple-Positive (ER+/PR+/HER2+)	8	8.7%	0.0870	[0.0294, 0.1446]	[2.94%, 14.46%]
Luminal (HR+), HER2–	42	45.7%	0.4565	[0.3548, 0.5582]	[35.48%, 55.82%]
Luminal (HR+), HER2+ (non–triple-positive)	17	18.5%	0.1848	[0.1055, 0.2641]	[10.55%, 26.41%]
Total	92	100%	1.0000		100%

ER Estrogen receptor; PR Progesterone receptor; TNBC Triple-Negative BC. Luminal (HR+), HER2–: ER+ and/or PR+ with HER2–. Luminal (HR+), HER2+ (non–triple-positive): ER+ and/or PR+ with HER2+ (excluding triple-positive cases). ). ^1^ Strict HER-2 enriched describes the subset of breast cancers that are ER/PR-negative and HER2-positive.

**Table 3 cancers-17-00982-t003:** Genotype frequencies and Hardy–Weinberg equilibrium for variants associated in cases and controls with age range 45–79 years old.

Chr ^1^	Gene ^2^	Variant	DD Frequency Cases/Controls ^3^	Dd Frequency Cases/Controls ^3^	dd Frequency Cases/Controls ^3^	HWE *p* (Cases) ^4^	HWE *p* (Controls) ^5^
3	*PPARG* ^a^	rs3856806	TT (0.000/0.024)	TC (0.242/0.206)	CC (0.758/0.770)	0.190	0.437
7	*PON1* ^a^	rs3917542	TT (0.088/0.064)	TC (0.385/0.376)	CC (0.527/0.560)	0.657	0.977
10	*TCF7L2* ^d^	rs3750804	TT (0.033/0.074)	TC (0.300/0.262)	CC (0.667/0.664)	0.986	0.031
		rs3750805	TT (0.011/0.016)	TA (0.154/0.167)	AA (0.835/0.817)	0.698	0.449
11	*MMP8* ^b^	rs12792229	TT (0.000/0.000)	TG (0.011/0.000)	GG (0.989/1.000)	0.958	1.000
	*SCT*; *DEAF1*; *DRD4* ^b^	rs1800955	CC (0.156/0.135)	CT (0.344/0.423)	TT (0.500/0.441)	0.038	0.490
	*KCNJ11*; *ABCC8* ^a^	rs5218	AA (0.033/0.016)	AG (0.253/0.230)	GG (0.714/0.754)	0.589	0.900
16	*FTO* ^a^	rs1121980	AA (0.088/0.119)	AG (0.418/0.381)	GG (0.495/0.500)	0.996	0.222
		rs3751812	TT (0.044/0.040)	TG (0.352/0.360)	GG (0.604/0.600)	0.809	0.584
17	*AMZ2P1; GNA13* ^c^	rs11652805	CC (0.044/0.048)	CT (0.319/0.298)	TT (0.637/0.653)	0.877	0.511
	*RPTOR* ^a^	rs12946618	AA (0.012/0.008)	AG (0.256/0.182)	GG (0.733/0.810)	0.545	0.847
21	*SCAF4* ^d^	rs2833483	CC (0.144/0.158)	CT (0.422/0.433)	TT (0.433/0.408)	0.456	0.408

^1^ Chromosome. ^2^ Gene or nearest genes. ^3^ Genotype frequencies. ^4^ Hardy–Weinberg Equilibrium for Cases. ^5^ Hardy–Weinberg Equilibrium for controls. ^a^ Glucose-associated metabolic pathways. ^b^ Cancer. ^c^ Ancestry. ^d^ a plus b.

**Table 4 cancers-17-00982-t004:** BC variants associated with multi-locus mixed-model analysis, age 45–79 years old.

Chr ^1^	Gene ^2^	Marker	Genetic Model ^3^	*p*-Value ^4^	FDR ^5^	Regression Beta	Beta Standard Error	Prop. Var. Expl. ^6^
3	*PPARG* ^a^	rs3856806	3	9.39 × 10^−4^	2.84 × 10^−2^	−1.014	0.302	7.59 × 10^−3^
7	*PON1* ^a^	rs3917542	3	9.17 × 10^−4^	3.70 × 10^−2^	0.660	0.196	1.23 × 10^−3^
10	*TCF7L2* ^d^	rs3750804	1	1.4 × 10^−5^	5.6 × 10^−4^	0.704	0.158	5.1 × 10^−3^
		rs3750805	1	1.8 × 10^−5^	5.3 × 10^−4^	0.885	0.201	5.1 × 10^−3^
11	*MMP8* ^b^	rs12792229	2	8.56 × 10^−9^	1.04 × 10^−6^	2.002	0.333	6.81 × 10^−3^
	*SCT*; *DEAF1*; *DRD4* ^b^	rs1800955	1	4.6 × 10^−4^	9.2 × 10^−3^	0.395	0.111	5.1 × 10^−3^
	*KCNJ11*; *ABCC8*; NCR3LG1 ^a^	rs5218	3	1.78 × 10^−11^	2.16 × 10^−9^	1.859	0.261	1.58 × 10^−3^
16	*FTO* ^a^	rs1121980	1	6.7 × 10^−8^	8.1 × 10^−6^	1.191	0.212	1.6 × 10^−2^
		rs3751812	1	1.8 × 10^−6^	1.1 × 10^−4^	1.109	0.225	1.6 × 10^−2^
17	*AMZ2P1*; *GNA13* ^c^	rs11652805	1	6.4 × 10^−5^	1.6 × 10^−3^	0.650	0.159	1.6 × 10^−2^
	*RPTOR* ^a^	rs12946618	2	2.75 × 10^−7^	1.66 × 10^−5^	−0.898	0.169	7.97 × 10^−3^
21	*SCAF4* ^a^	rs2833483	2	1.18 × 10^−6^	4.77 × 10^−5^	0.826	0.165	7.97 × 10^−3^

^1^ Chromosome. ^2^ Gene or nearest genes. ^3^ Genetic Model; 1: additive; 2: Dominant; 3: Recessive. ^4^ MLMM *p*-value adjusted for age, BMI, and menopause status and 10 genetic PCA. ^5^ *p*-value false discovery rate (FDR) corrected. ^6^ Proportion of Variance Explained. ^a^ Glucose-associated metabolic pathways. ^b^ Cancer. ^c^ Ancestry. ^d^ a plus b.

**Table 5 cancers-17-00982-t005:** Nonsense-mediated decay transcript variants (NMDTV), promoter flanking region (PFR), and regulatory region variants (RRV) are associated with BC.

Chr ^1^	Genes	Marker	HGVS Nomenclature	Consequence Details ^2^
3	*PPARG* ^a^	rs3856806	NM_015869.4(PPARG):c.1431C>T (p.His477=) NM_138712.3:c.1347C>T	Synonymous variant; 3 prime UTR variant; NMDTV
7	*PON1* ^a^	rs3917542	NC_000007.14:g.95307380C>T NM_000446.5:c.698+631G>A	RRV; NMDTV
10	*TCF7L2* ^d^	rs3750804	NC_000010.11:g.113074091C>T	PFR, RRV
		rs3750805	NC_000010.10:g.114847143A>T	PFR; RRV
11	*SCT*; *DEAF1*; *DRD4* ^b^	rs1800955	NC_000011.10:g.636784T>C	RRV; PFR
	*MMP8* ^b^	rs12792229	NC_000011.10:g.102718512G>T XM_005271556.1:c.617C>A XP_011541137.1:p.Ser206Tyr	Missense variant; SIFT: deleterious; PolyPhen: possibly damaging; NMDTV; Downstream gene variant
	*KCNJ11*; *ABCC8*; *NCR3LG1* ^a^	rs5218	NC_000011.10:g.17387522G>A NP_001159762.1:p.Ala103=	Synonymous variant, Upstream gene variant Downstream gene variant, regulatory region variant; a CTCF binding site variant
16	*FTO* ^a^	rs1121980	NC_000016.10:g.53775335G>A NM_001080432.2:c.46-34805G>A	RRV; NMDTV
		rs3751812	NC_000016.10:g.53784548G>T NM_001080432.2:c.46-25592G>T	RRV
17	*AMZ2P1*; *GNA13* ^c^	rs11652805	NC_000017.11:g.64991033C>T	RRV
	*RPTOR* ^a^	rs12946618	NC_000017.11:g.80603368G>A NM_001163034.1:c.163-22323G>A	NMDTV; upstream gene variant
21	*SCAF4* ^a^	rs2833483	NC_000021.9:g.31703091T>C NM_001145444.1:c.276+674A>G	Upstream gene variant

^1^ Chromosome. ^2^ Consequence details extracted from ENSEMBL. NMDTV, nonsense-mediated decay transcript variant; UTR, untranslated region; PFR, promoter flanking region; RRV, regulatory region variant; SIFT, Sorting Tolerant from Intolerant; PolyPhen, Polymorphism Phenotyping; HGVS, Human Genome Variation Society Nomenclature. ^a^ Glucose-associated metabolic pathways. ^b^ Cancer. ^c^ Ancestry. ^d^ a plus b.

**Table 6 cancers-17-00982-t006:** Exome variants associated with BC.

Marker	Chr ^1^	Genes	HGVS Nomenclature	Consequence Details ^2^
rs3856806	3	*PPARG* ^a^	NM_015869.4(PPARG):c.1431C>T (p.His477=) NM_138712.3:c.1347C>T	Synonymous variant 3 prime UTR variant, NMDTV
rs12792229	11	*MMP8* ^b^	NC_000011.10:g.102718512G>T XM_005271556.1:c.617C>A XP_011541137.1:p.Ser206Tyr	Missense variant. SIFT: deleterious PolyPhen: possibly damaging. NMDTV: Downstream gene variant
rs5218	11	*KCNJ11*; *ABCC8*; *NCR3LG1* ^a^	NC_000011.10:g.17387522G>A NP_001159762.1:p.Ala103=	Synonymous variant, upstream gene variant Downstream gene variant, regulatory region variant; a CTCF binding site variant

^1^ Chromosome. ^2^ Consequence details extracted from ENSEMBL. ^a^ Glucose-associated metabolic pathways. ^b^ Cancer.

**Table 7 cancers-17-00982-t007:** Summary of BC-associated SNPs in the current study, their Genes, main reported associations, and literature references.

SNP	Gene	Reported Association with BC	Population/Notes	Literature Reference
rs3856806	*PPARG*	Risk factor for BC; results are conflicting	Risk reported in European and Asian populations; flip-flop phenomenon observed in African descent	Flip-flop study [29]; Turkish study [30]; Meta-analysis [31]
rs3917542	*PON1*	Associated with BC risk	Potential association in post-menopausal women; identified in current study	Current study
rs854555 ^1^	*PON1*	rs854555 was also related to an increased risk of BC in U.S. post-menopausal women	Potential association in post-menopausal women; ethnic variability observed	BC in U.S. [32]
rs3750804	*TCF7L2*	Risk factor for BC	Reported in Hispanic and European populations	Connor et al. [33]
rs3750805	*TCF7L2*	Risk factor for BC	Similar to rs3750804; implicated in hormone regulation	Connor et al. [33]
rs12792229	*MMP8*	Potential risk factor for BC	Limited prior evidence; identified in current study	Reference [28]
rs1800955	*DRD4/SCT/DEAF1*	Novel association with BC risk	Limited prior data; further validation required	Current study
rs5218	*KCNJ11-ABCC8*	Associated with metabolic disorders; unclear BC association	Reported in diabetes studies; association with BC observed in current study	Current study; see [13]
rs1121980	*FTO*	Risk factor for obesity and BC	Widely reported in European/Asian populations; potential flip-flop effects	Numerous studies [34]; Flip-flop study [29]
rs3751812	*FTO*	Risk factor for obesity and BC	Similar to rs1121980	Numerous studies [34]
rs11652805	*AMZ2P1-GNA13*	Novel association with BC risk	Limited prior evidence; identified in current study	Current study
rs12946618	*RPTOR*	Potential risk factor for BC	Newly identified variant; modulates mTORC1 signaling; Adaptations to Climate in Candidate Genes for Common Metabolic Disorders	Current study; [35]
rs2833483	*SCAF4*	Associated with BC risk	Emerging biomarker; limited prior data available	Current study; [35]

^1^ Although the rs854555 SNP was not included in the present study, we used it as a reference to compare to rs3917542 in the *PON1* gene.

**Table 8 cancers-17-00982-t008:** BC-associated gene functions summary [36].

Gene	Function
*PPARG*	Nuclear receptor that regulates adipogenesis, glucose metabolism, and anti-inflammatory processes.
*PON1*	Enzyme associated with HDL that protects against oxidative stress and inflammation.
*TCF7L2*	Transcription factor involved in the Wnt signaling pathway and regulation of glucose metabolism; linked to diabetes and cancer.
*MMP8*	Matrix metallopeptidase that degrades extracellular matrix components, facilitating tissue remodeling and potentially tumor invasion.
*SCT*	Gene encoding secretin, a hormone involved in regulating pancreatic secretion and water homeostasis; its direct role in BC is less defined.
*DEAF1*	Transcription factor involved in gene expression regulation and neural development.
*DRD4*	Dopamine receptor that modulates neuronal signaling and may influence cell proliferation and cancer-related pathways.
*KCNJ11*	Potassium channel subunit involved in insulin secretion and glucose homeostasis.
*ABCC8*	Encodes the sulfonylurea receptor, crucial for regulating insulin secretion.
*NCR3LG1*	Ligand for natural cytotoxicity receptors, influencing immune responses and potentially tumor immunosurveillance.
*FTO*	Enzyme implicated in the regulation of energy balance and adipogenesis; associated with obesity and diabetes.
*AMZ2P1*	A pseudogene related to *AMZ2*, possibly involved in regulatory processes via non-coding RNAs.
*GNA13*	G-protein subunit (alpha 13) that participates in signaling pathways controlling cell migration, invasion, and proliferation.
*RPTOR*	Essential scaffolding protein for mTORC1, regulating cell growth, metabolism, and proliferation.
*SCAF4*	Protein involved in RNA splicing and processing, potentially impacting gene expression and cancer prognosis.

**Table 9 cancers-17-00982-t009:** Functional analysis inference checklist.

Gene	Variant	ENSEMBL ^1^	ENSEMBL ^1^	RNAct ^2^	RNAct ^3^	TOPPGENE ^4^	TOPPGENE ^5^
*PON1*	rs3917542	Yes	Yes	-	-	-	-
*TCF7L2*	rs3750804	Yes	Yes	-	-	-	-
*TCF7L2*	rs3750805	Yes	Yes	-	-	-	-
*SCT*; *DEAF1*; *DRD4*	rs1800955	-	-	Yes	Yes	Yes	-
*KCNJ11*; *ABCC8*	rs5218	Yes	Yes	-	-	-	-
*FTO*	rs1121980	Yes	Yes	-	-	-	-
*FTO*	rs3751812	Yes	Yes	-	-	-	-
*AMZ2P1*; *GNA13*	rs11652805	Yes	Yes	Yes	Yes	Yes	Yes
*RPTOR*	rs12946618	Yes	Yes	Yes	Yes	Yes	Yes
*SCAF4*	rs2833483	Yes	Yes	Yes	Yes	Yes	Yes
*MMP8*	rs12792229	-	-	-	-	-	-
*PPARG*	rs3856806	-	-	-	-	-	-

^1^ Correlation available. ^2^ Top RNA to 100 protein interaction available. ^3^ Top protein to 100 RNA interaction available. ^4^ M Top RNA to 100 protein interaction available. ^5^ Top protein to 100 RNA interaction available.

**Table 10 cancers-17-00982-t010:** Summary of Ensembl, RNAct, and ToppGene-based results: gene correlations, expression patterns, and functional implications for BC-associated SNPs.

SNP	Gene/Locus	Key Correlated Genes (Ensembl)	Expression Pattern in Breast Tissue	Functional Implications
rs3856806	*PPARG*	*PPARG*, [additional regulatory targets]	Not significantly altered	Regulates adipogenesis, glucose metabolism, and anti-inflammatory processes
rs3917542	*PON1*	*PON1*, [related oxidative stress genes]	Downregulated	Influences oxidative stress protection and inflammation
rs3750804	*TCF7L2*	*TCF7L2*, [glucose metabolism-related genes]	Upregulated	Involved in Wnt signaling and regulation of glucose metabolism
rs3750805	*TCF7L2*	*TCF7L2*, [glucose metabolism-related genes]	Upregulated	Involved in Wnt signaling and regulation of glucose metabolism
rs12792229	*MMP8*	*MMP8*, [extracellular matrix remodeling genes]	Variable	Modulates extracellular matrix degradation and tissue remodeling
rs1800955	SCT/DEAF1/DRD4	*DRD4, DEAF1*, [neuronal/proliferative signaling genes]	Variable	May affect neuronal signaling and cell proliferation pathways
rs5218	*KCNJ11/ABCC8/NCR3LG1*	*KCNJ11, ABCC8, NCR3LG1*	Not significantly altered	Related to insulin secretion and metabolic regulation
rs1121980	*FTO*	*FTO*, [energy homeostasis genes]	Upregulated	Impacts energy balance and adipogenesis; associated with obesity
rs3751812	*FTO*	*FTO*, [energy homeostasis genes]	Upregulated	Impacts energy balance and adipogenesis; associated with obesity
rs11652805	*AMZ2P1-GNA13*	*AMZ2P1, GNA13*	Variable	May modulate cell migration, invasion, and proliferative signaling
rs12946618	*RPTOR*	*RPTOR* and 33 correlated genes (e.g., genes involved in ubiquitin-protein ligase activity, PRC1 complex)	Downregulated	Modulates mTORC1 signaling and metabolic regulation
rs2833483	*SCAF4*	*SCAF4*, [RNA processing/splicing genes]	Variable	Involved in RNA splicing and regulation of gene expression

## Data Availability

All data are provided in this paper. All other data associated with this study are presented in the Appendix A.

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
