# Peer review of "Genetic Insights into Breast Cancer in Northeastern Mexico: Unveiling Gene–Environment Interactions and Their Links to Obesity and Metabolic Diseases"

_cancers, 2025, doi:10.3390/cancers17060982_

Round 1
Reviewer 1 Report
Comments and Suggestions for Authors
The manuscript explored the association of specific genetic variants with breast cancer (BC) risk, menopause, and body mass index (BMI) in a population from Northeast Mexico. They suggested that 12 gene variants were significantly associated with BC. Considering that I could not evaluate the Supplementary 1 file, I cannot express my opinion about the article exactly. There are number of major revision, formal and scientific aspects that should be addressed.
- In the case of the control group mentioned, you sampled healthy individuals, if this claim is true, how did these individuals agree to be sampled?
- Why did you select the SNP hsa-miR-196a2 because this microRNA is involved in Acute lymphoblastic leukemia (ALL)?
- The patient's demographic information in Table 1 needs to be completed such as smoking, alcohol, sedentary lifestyle, and etc.
- The information in Table 2 seems to be incorrect.
- It is necessary to state the limitations that existed in carrying out the project.
- I was unable to review Supplementary 1. Apparently, access has been restricted. I need to restore my access to the files.
- It seems that rs12946618 RPTOR does not play a role in breast cancer. It is necessary to add the necessary explanations to the text of the article about its association with this cancer.
- It is necessary to draw statistical graphs to prove the correlation of these SNPs with breast cancer.
Author Response
Please see the attachment; we provide a point-by-point response to reviewer 1 comments as a Word file.
Response to Reviewer 1 Comments
- Summary
Thank you very much for taking the time to review this manuscript. Please find the detailed responses below and the corresponding revisions/corrections highlighted/in track changes in the re-submitted files.
- Consent in Controls: We clarified that the term "subjects" should be used, as patients and controls signed informed consent forms before participation.
- Selection of hsa‐miR‐196a2: We explained that although hsa‐miR‐196a2 is implicated in various cancers (including acute lymphoblastic leukemia), it has also been associated with breast cancer in multiple studies. We detailed its biological role in regulating gene expression, cell proliferation, and apoptosis (factors relevant to breast tumor biology) and emphasized the need for further functional studies and replication in diverse populations.
- Demographic Data: We acknowledged the correct reviewer's suggestion to include lifestyle variables (e.g., smoking, alcohol, sedentary behavior) in Table 1. However, we noted that limited information was available from medical records and questionnaires due to incomplete data and privacy concerns. We regret the omission and plan to address this in future studies.
- Correction of Table 2: We have carefully reviewed and corrected Table 2 to ensure accurate data presentation, and we appreciate the reviewer's attention to detail.
- Study Limitations: We expanded the Discussion of our study's limitations. These include the modest sample size and retrospective design, the geographical and ethnic specificity of the sample, the limited genetic scope (focusing on 121 SNPs), incomplete lifestyle data, and the lack of functional validation for the identified associations.
- Access to Supplementary Material: We noted that if the reviewer cannot access Supplementary File 1, they should communicate this issue to the editor.
- Role of rs12946618 (RPTOR): We provided a deeper explanation for rs12946618 association result despite limited prior data linking it to breast cancer. Our analysis indicates that this regulatory variant may impact RPTOR expression and, consequently, mTORC1 signaling, a pathway involved in oncogenesis. We further detailed gene expression correlations and proposed mechanisms, emphasizing the need for future functional studies and extensive replication studies.
- Statistical Graphs: We confirmed that our manuscript includes statistical evidence. Figure 2 presents Q‑Q plots for the additive, dominant, and recessive genetic models, while Figure 3 provides a Manhattan plot with a clear significance cut-off. These figures validate the correlation between the SNPs and breast cancer risk.
We believe these responses address the reviewer's concerns and have strengthened our manuscript accordingly.
Point-by-point response to Comments and Suggestions for Authors
Comments 1: In the case of the control group mentioned, you sampled healthy individuals, if this claim is true, how did these individuals agree to be sampled?
Response 1: Indeed, it is an error, it should have said subjects, since both patients and controls signed an informed consent.
Comments 2: Why did you select the SNP hsa-miR-196a2 because this microRNA is involved in Acute lymphoblastic leukemia (ALL)?.
Response 2: The hsa-miR-196a2 has been associated with various types of cancer in 644 articles, in 46 of 644 has reported association with BC.
Importance of hsa-miR-196a2 in the biology of breast cancer.
-The microRNA hsa-miR-196a2 is expressed in breast tissue (RNAseq).
-Gene regulation: miR-196a2 regulates various genes related to proliferation, apoptosis, and cell differentiation. It regulates genes in signaling pathways such as TGF-β, Wnt/β-catenin, and others associated with proliferation, differentiation, and cell migration, all of which are critical in breast tumor biology.
-Overexpression and tumor progression: Several studies have reported that overexpression of miR-196a2 is associated with various types of cancer (including lung, gastric, colorectal, and breast cancer). However, the extent of its involvement may vary depending on the cell lineage and the genetic background of the population studied.
-Additionally, miR-196a2 and the SNP rs11614913 have been associated with phenotypes such as BMI-adjusted waist circumference and BMI-adjusted waist-hip ratio and breast cancer. The association with overweight and obesity phenotypes (indirectly related to insulin resistance and diabetes) underscores the importance of evaluating the interaction with other genes, environmental risk factors (such as insulin resistance, overweight, and obesity), and the epigenetic context regarding its association with cancer.
-Because of these inconclusive previous studies, it is necessary to validate the results concerning rs11614913 in breast cancer risk; it is essential to:
-
- More replication studies are required, including large samples, diverse ethnicities, and populations with complex genetic structures.
- Conduct in-depth functional studies: Beyond statistical association, functional (in vitro and in vivo) research is needed to explain how this polymorphism modulates miR-196a2 activity, its targets, and the tumor phenotype.
- Examine interactions with other polymorphisms and risk factors: Breast cancer is multifactorial. The contribution of a single SNP in a single miRNA is modest or depends on interactions with other genes (epistasis) and epigenetic and environmental factors. Therefore, polygenic models and gene-environment (multifactorial) interaction studies are crucial to elucidate the actual relevance of miR-196a2.
Additionally, we provide a list of publications on microRNA miR-196a-2 and breast cancer.
|
Year |
PMID |
Title |
Author(s) |
|
2008 |
ATM in breast cancer susceptibility - results of a pooled analysis of case-control mutation screening data |
Babikyan, D.; Lesueur, F.; Voegele, C.; Vallee, et al |
|
|
2008 |
Genetic variants in fibroblast growth factor receptor 2 (FGFR2) contribute to susceptibility of breast cancer in Chinese women |
Shen, H.; Hu, Z.; Chen, J.; Tian, et al |
|
|
2009 |
18634034 |
Common genetic variants in pre-microRNAs were associated with increased risk of breast cancer in Chinese women. |
Hu Z, Liang J, Wang Z, Tian T, et al |
|
2009 |
19567675 |
microRNA miR-196a-2 and breast cancer: a genetic and epigenetic association study and functional analysis. |
Hoffman AE, Zheng T, Yi C, Leaderer D, et al |
|
2010 |
19847796 |
Evaluation of SNPs in miR-146a, miR196a2 and miR-499 as low-penetrance alleles in German and Italian familial breast cancer cases. |
Catucci I, Yang R, Verderio P, Pizzamiglio S, et al |
|
2011 |
20640596 |
The association between two polymorphisms in pre-miRNAs and breast cancer risk: a meta-analysis. |
Gao LB, Bai P, Pan XM, Jia J, et al |
|
2011 |
21962133 |
Single nucleotide polymorphism in hsa-mir-196a-2 and breast cancer risk: a case control study. |
Jedlinski DJ, Gabrovska PN, Weinstein SR, Smith RA, et al |
|
2012 |
22074121 |
Associations of miRNA polymorphisms and female physiological characteristics with breast cancer risk in Chinese population. |
Zhang M, Jin M, Yu Y, Zhang S, et al |
|
2012 |
23228090 |
Evaluation of single nucleotide polymorphisms in microRNAs (hsa-miR-196a2 rs11614913 C/T) from Brazilian women with breast cancer. |
Linhares JJ, Azevedo M, Siufi AA, de Carvalho CV, et al |
|
2012 |
22363415 |
Differential expression profile and genetic variants of microRNAs sequences in breast cancer patients. |
Alshatwi AA, Shafi G, Hasan TN, Syed NA, et al |
|
2012 |
22363684 |
Increased risk of breast cancer associated with CC genotype of Has-miR-146a Rs2910164 polymorphism in Europeans. |
Lian H, Wang L, Zhang J. |
|
2012 |
22586447 |
Polymorphism rs4919510:C>G in mature sequence of human microRNA-608 contributes to the risk of HER2-positive breast cancer but not other subtypes. |
Huang AJ, Yu KD, Li J, Fan L, et al |
|
2013 |
24062209 |
Genetic variants in microRNAs and breast cancer risk in African American and European American women. |
Yao S, Graham K, Shen J, Campbell LE, et al |
|
2013 |
24039706 |
The associations of single nucleotide polymorphisms in miR-146a, miR-196a and miR-499 with breast cancer susceptibility. |
Wang PY, Gao ZH, Jiang ZH, Li XX, et al |
|
2013 |
23555923 |
There is no association between microRNA gene polymorphisms and risk of triple negative breast cancer in a Chinese Han population. |
Ma F, Zhang P, Lin D, Yu D, et al |
|
2013 |
24260062 |
Higher expression of circulating miR-182 as a novel biomarker for breast cancer. |
Wang PY, Gong HT, Li BF, Lv CL, et al |
|
2014 |
23982873 |
Ethnicity modifies the association between functional microRNA polymorphisms and breast cancer risk: a HuGE meta-analysis. |
Chen QH, Wang QB, Zhang B. |
|
2014 |
25374621 |
Common genetic variants in pre-microRNAs and risk of breast cancer in the North Indian population. |
Bansal C, Sharma KL, Misra S, Srivastava AN, et al |
|
2014 |
24829853 |
Micro-RNAs as clinical biomarkers and therapeutic targets in breast cancer: Quo vadis? |
Christodoulatos GS, Dalamaga M. |
|
2014 |
24521023 |
hsa-mir-499 rs3746444 gene polymorphism is associated with susceptibility to breast cancer in an Iranian population. |
Omrani M, Hashemi M, Eskandari-Nasab E, Hasani SS, et al |
|
2014 |
24922658 |
Genetic polymorphism of miR-196a as a prognostic biomarker for early breast cancer. |
Lee SJ, Seo JW, Chae YS, Kim JG, et al |
|
2015 |
26577090 |
Genetic association analysis of miRNA SNPs implicates MIR145 in breast cancer susceptibility. |
Chacon-Cortes D, Smith RA, Haupt LM, Lea RA, et al |
|
2015 |
25483824 |
Five common functional polymorphisms in microRNAs (rs2910164, rs2292832, rs11614913, rs3746444, rs895819) and the susceptibility to breast cancer: evidence from 8361 cancer cases and 8504 controls. |
Dai ZJ, Shao YP, Wang XJ, Xu D, et al |
|
2015 |
25759599 |
Critical analysis of the potential for microRNA biomarkers in breast cancer management. |
Graveel CR, Calderone HM, Westerhuis JJ, Winn ME, et al |
|
2015 |
26125831 |
Associations of miRNA polymorphisms and expression levels with breast cancer risk in the Chinese population. |
Qi P, Wang L, Zhou B, Yao WJ, et al |
|
2016 |
26886638 |
The Associations of Single Nucleotide Polymorphisms in miR196a2, miR-499, and miR-608 With Breast Cancer Susceptibility: A STROBE-Compliant Observational Study. |
Dai ZM, Kang HF, Zhang WG, Li HB, et al |
|
2016 |
27421647 |
Association of single nucleotide polymorphisms in Pre-miR-27a, Pre-miR-196a2, Pre-miR-423, miR-608 and Pre-miR-618 with breast cancer susceptibility in a South American population. |
Morales S, Gulppi F, Gonzalez-Hormazabal P, Fernandez-Ramires R, et al |
|
2016 |
26710106 |
Somatic Mutation of the SNP rs11614913 and Its Association with Increased MIR 196A2 Expression in Breast Cancer. |
Zhao H, Xu J, Zhao D, Geng M, et al |
|
2016 |
27105503 |
ANXA1 inhibits miRNA-196a in a negative feedback loop through NF-kB and c-Myc to reduce breast cancer proliferation. |
Yuan Y, Anbalagan D, Lee LH, Samy RP, et al |
|
2017 |
28950676 |
Association between Microrna 146a and Microrna 196a2 Genes Polymorphism and Breast Cancer Risk in North Indian Women |
Bodal VK, Sangwan S, Bal MS, Kaur M, et al |
|
2017 |
27880723 |
Association between three functional microRNA polymorphisms (miR-499 rs3746444, miR-196a rs11614913 and miR-146a rs2910164) and breast cancer risk: a meta-analysis. |
Zhang H, Zhang Y, Yan W, Wang W, et al |
|
2017 |
28978158 |
Meta-analysis of the association between three microRNA polymorphisms and breast cancer susceptibility. |
Mu K, Wu ZZ, Yu JP, Guo W, et al |
|
2018 |
29521182 |
Effects of miR-27a, miR-196a2 and miR-146a polymorphisms on the risk of breast cancer. |
Mashayekhi S, Saeidi Saedi H, Salehi Z, Soltanipour S, et al |
|
2018 |
29782194 |
Five Common Functional Polymorphisms in microRNAs and Susceptibility to Breast Cancer: An Updated Meta-Analysis. |
Wu J, Wang Y, Shang L, Qi L, et al |
|
2018 |
29965793 |
miRNA 196a2(rs11614913)Â &Â 146a(rs2910164) polymorphisms & breast cancer risk for women in an Iranian population. |
Nejati-Azar A, Alivand MR. |
|
2018 |
30135399 |
Genetic Variants in pre-miR-146a, pre-miR-499, pre-miR-125a, pre-miR-605, and pri-miR-182 Are Associated with Breast Cancer Susceptibility in a South American Population. |
Morales S, De Mayo T, Gulppi FA, Gonzalez-Hormazabal P, et al |
|
2019 |
30781715 |
Genetic Epidemiology of Breast Cancer in Latin America. |
Zavala VA, Serrano-Gomez SJ, Dutil J, Fejerman L. |
|
2019 |
31777500 |
Genetic polymorphism of miRNA-196a and its target gene annexin-A1 expression based on ethnicity in Pakistani female breast cancer patients. |
Rahim A, Afzal M, Naveed AK. |
|
2019 |
30783203 |
MicroRNA-196a is regulated by ER and is a prognostic biomarker in ER+ breast cancer. |
Milevskiy MJG, Gujral U, Del Lama Marques C, Stone A, et al |
|
2019 |
30897768 |
Breast Cancer and miR-SNPs: The Importance of miR Germ-Line Genetics. |
Malhotra P, Read GH, Weidhaas JB. |
|
2020 |
32320336 |
Predictive role of single nucleotide polymorphism (rs11614913) in the development of breast cancer in Pakistani population. |
Ahmad M, Shah AA. |
|
2020 |
33110366 |
Single nucleotide polymorphisms in microRNAs action as biomarkers for breast cancer. |
Nguyen TTN, Tran MTH, Nguyen VTL, Nguyen UDP, et al |
|
2021 |
34454612 |
Association between single-nucleotide polymorphisms in miRNA and breast cancer risk: an updated review. |
Arancibia T, Morales-Pison S, Maldonado E, Jara L. |
|
2022 |
35971775 |
In silico identification and in vitro expression analysis of breast cancer-related m6A-SNPs. |
Kleinbielen T, Olasagasti F, Azcarate D, Beristain E, et al |
|
2023 |
36831624 |
Potential Impact of PI3K-AKT Signaling Pathway Genes, KLF-14, MDM4, miRNAs 27a, miRNA-196a Genetic Alterations in the Predisposition and Progression of Breast Cancer Patients. |
Alzahrani OR, Mir R, Alatwi HE, Hawsawi YM, et al |
|
2023 |
37899898 |
Prospective Functions of miRNA Variants May Predict Breast Cancer Among Saudi Females. |
Ekram SN, Alghamdi G, Elhawary AN, Sembawa HA, et al |
Based on the above, we established the rationale for its analysis. However, the SNP in question did not yield significant results, so no information will be added to the revised document.
Comments 3: The patient's demographic information in Table 1 needs to be completed such as smoking, alcohol, sedentary lifestyle, and etc.
Response 3:
We appreciate the reviewer's suggestion regarding the inclusion of additional demographic and lifestyle information (e.g., smoking status, alcohol consumption, and sedentary behavior) in Table 1. Unfortunately, there were several challenges that prevented us from collecting and presenting those specific variables in our dataset:
Limited Availability in the Medical Records: Limited Availability in the Medical Records: Unfortunately, not all participating institutions have consistently registered detailed lifestyle factors. In many cases, the health records lacked full or accurate entries for smoking habits, alcohol use, and physical activity.
Incomplete Patient Questionnaires: Where self-reported data were intended (subjective and lack of scientific accuracy), many participants either did not complete or only partially completed the lifestyle questionnaires. This incomplete data reduced the feasibility of accurately quantifying these factors without introducing bias.
Ethical and Privacy Considerations: Some participants opted not to disclose personal habits. We could not compel them to provide these details out of respect for patient confidentiality and in adherence to institutional review board (IRB) guidelines.
Given these constraints, we focused on the demographic and clinical variables for which complete, reliable information was obtained.
We deeply regret that we cannot include these factors in the present study. We will still work to address these logistic problems in future studies.
Comments 4: The information in Table 2 seems to be incorrect.
Response 4:
We appreciate their invaluable attention to detail and suggestions, which have helped improve the clarity and precision of our manuscript. Please find the revised Table 2 in the updated version of our submission.
If there are any further concerns or additional clarifications needed, we are happy to address them.
Comments 5: It is necessary to state the limitations that existed in carrying out the project.
Response 5:
Response to Reviewer Comment Regarding Study Limitations
We appreciate emphasizing the need to state the limitations of our study more explicitly. We have now included the following points to clarify the main constraints encountered during the project:
-
- Sample Size and Study Design:
-
-
- The retrospective, case-control design and the relatively modest sample size (91 cases, 126 controls) limit the statistical power of our findings and may reduce the ability to detect smaller effect sizes. However, the population structure is homogeneous. The sample size is small, however the simultaneous analysis of 121 SNPs in both cases and controls provide valuable insights into the genetic composition of the studied population. Additionally, as this was not a longitudinal study, we could not assess causality or disease progression.
-
-
- Geographical and Ethnic Specificity:
-
-
- All participants were recruited from a specific region in Northeastern Mexico, which may limit the generalizability of our conclusions to other populations. Although we employed ancestry-informative markers (AIMs) to address population stratification, our findings may reflect unique genetic and environmental factors specific to this region.
-
-
- Restricted Genetic Scope:
-
-
- We focused on 121 SNPs previously implicated in breast cancer, metabolic disorders, and ancestry. While this targeted approach allowed for in-depth analysis of certain variants, it did not capture the entire spectrum of possible genetic risk factors.
-
-
- Incomplete Lifestyle and Environmental Data:
-
-
- Although we considered age, menopausal status, and BMI as cofactors, detailed information on diet, physical activity, alcohol consumption, and smoking was not systematically available for all participants. Leaked information on diet, physical activity, alcohol consumption, and smoking limits our ability to assess fully the impact of gene-environment interactions on breast cancer risk. We are still assessing financial support for NGS studies in BC gene-environment interactions.
-
-
- Functional Validation:
-
-
- Our study centered on statistical associations; thus, functional assays to confirm or elucidate the biological impact of identified risk variants were beyond this project's scope. Further molecular and mechanistic investigations are needed to validate the direct influence of these SNPs on breast cancer development. We are still assessing financial support for functional validation studies for these genes and variants.
-
We appreciate the reviewer's insight and have added a concise "Limitations" subsection in the revised manuscript to highlight these points explicitly. This acknowledgment will help contextualize the findings and guide future research aimed at overcoming these constraints.
Comments 6: I was unable to review Supplementary 1. Apparently, access has been restricted. I need to restore my access to the files.
Response 6:
We regret that you cannot review the additional material provided, perhaps you should comment to the editor about this.
Comments 7: It seems that rs12946618 RPTOR does not play a role in breast cancer. It is necessary to add the necessary explanations to the text of the article about its association with this cancer.
Response 7:
We are grateful for drawing attention to the need for further explanation regarding rs12946618 in RPTOR and its possible relationship to breast cancer.
As we stated in lines 514 and 515: "We found rs12946618 associated with BC. To our knowledge, no previous reports of this SNP and BC risk exist."
Lines 515 to 551, page 16, explain the implications of RPTOR in molecular regulatory mechanisms and insulin receptor signaling. The rs12946618 variant in RPTOR may lead to reduced RPTOR protein expression in both homozygous and heterozygous carriers, potentially affecting biological processes. The GTEx Project (https://gtexportal.org/home/) reports genes to be affected by the variant rs12946618; these genes are listed in the ENSEMBL database "Gene expression correlations." On page 16, lines 517-524, we report the bioinformatics analysis of 33 genes expressed and correlated to variant rs12946618 in several tissues:
"The Toppgene analysis of the 33 genes expressed correlated results in a molecular function in a single-stranded RNA binding, methylated histone binding, and cellular component PRC1 complex, a protein complex that includes a ubiquitin-protein ligase and enables ubiquitin-protein ligase activity (EMBL-EBI database, GO:0000151), the PcG protein complex; transcriptional repressors that regulate several crucial developmental and physiological processes [98]. The related pathways are Reactome regulation of PTEN gene transcription and Reactome sumoylation of DNA methylation proteins."
In context, the ubiquitin-protein ligase has a role in protein regulation/degradation (page 16, lines 517-524). The RNA-binding ubiquitin ligases are proteins that regulate RNA stability and link RNA-mediated mechanisms to protein ubiquitylation. They are involved in regulating mRNA processing, stability, and metabolism. DOI: 10.1042/BST0381621.
Below is the additional clarification we could be included in the supplementary manuscript:
Gene expression correlated to rs12946618 variant specific in Breast Mammary Tissue.
Downregulation: Sixteen genes were downregulated: RNF213-AS1, TBC1D16, EIF4A3, RPTOR, GAA, SLC38A10, NPTX1, CBX8, CARD14, BAIAP2-DT, BAIAP2, CHMP6, CCDC40, CBX2, BAHCC1, and TMEM105 in breast mammary tissue (GTEx project, ENSEMBL), and correlated to the rs12946618 variant. Molecular pathways: To determine the relevance of genes downregulated, the sixteen genes downregulated were analyzed in the ToppGene Suite (http://toppgene.cchmc.org). The Cellular Component Gene Ontology describes the location of parts of cells and the structures of the cell; this information is helpful in understanding where molecular processes occur with the sixteen genes downregulated. The Cellular Components associated (p-value FDR B&H < 0.005) were the PRC1 complex, nuclear ubiquitin ligase complex, PcG protein complex, neuron projection branch point, dendritic spine cytoplasm, autolysosome lumen, and heterochromatin are relevant to cancer development and progression. Key points: Many studies focus on the upregulation or hyperactivation of the referred molecular pathways for treatment strategies. On the other hand, the reduction of activity in these pathways can disrupt normal cellular functions and contribute to oncogenesis. The downregulation of these pathways disrupts normal cellular homeostasis, contributing to the loss of controlled gene expression, accumulation of damaged proteins, and genomic instability, all critical factors in cancer development.
Upregulation. On the other hand, TEPSIN has been upregulated in mammary tissue, correlated to variant rs12946618, and associated with the "AP-4 complex accessory subunit pepsin measurement disease". Gene ontology identifies it as a component of the "AP-4 adaptor complex cellular component" (AP4S1, TEPSIN, AP4E1, AP4M1, and AP4B1 gene).
AP-4 is a transcription factor involved in tumor biology and poor patient survival. It is overexpressed in tumors of various organs, including the lungs, stomach, and colon.
It is relevant that 13 of 15 upregulated genes were located in the chr17q25 cytoband (Ensembl 112 genes in cytogenetic band chr17q25), and 6 of 15 genes were identified as coexpressed on the Human Gene Set; "NIKOLSKY BREAST CANCER 17Q21 Q25 AMPLICON" (GSEA M15936). Genes within amplicon 17q21-q25 were identified in a copy number alterations study of 191 breast tumor samples.
Biological Significance of RPTOR
-
- RPTOR encodes a crucial scaffolding protein in the mammalian target of rapamycin complex 1 (mTORC1). This complex regulates cell growth, proliferation, and metabolism—key processes that can be dysregulated in breast cancer.
- Prior studies have shown that hyperactivation of mTORC1 promotes tumorigenesis, therapy resistance, and altered cellular metabolism. Although the literature on rs12946618 is explicitly limited, genetic variants within RPTOR can modulate its expression or function, thereby influencing mTORC1 signaling.
Association with Breast Cancer in Our Cohort
-
- Our multi-locus mixed-model analysis (MLMM) identified rs12946618 as significantly associated with BC risk in the Northeast Mexico population. Notably, RPTOR variants have also been linked to other malignancies, suggesting a broader tumorigenic role for this gene.
- While we acknowledge that the functional impact of rs12946618 on RPTOR expression or activity is not definitively established, in silico analyses indicated that this SNP overlaps with regulatory elements (promoter flanking or enhancer-like regions) and nonsense-mediated decay transcripts that could affect RPTOR expression and, consequently, mTORC1 activity.
Possible Mechanisms and Future Directions
-
- Dysregulation of mTORC1 may increase cell proliferation, survival signals, and metabolic reprogramming processes often observed in breast cancer pathogenesis.
- Further molecular and functional studies (e.g., in vitro assays, gene expression analysis, and reporter assays) are needed to clarify whether rs12946618 impacts RPTOR transcription or mRNA stability.
- Larger, multi-ethnic replication studies can help validate our findings and pinpoint whether this variant has a population-specific effect or represents a broader marker of breast cancer susceptibility.
We hope this additional information addresses concerns and underscores the potential relevance of RPTOR polymorphisms to breast cancer biology in our study cohort.
In summary, this is the first time the rs12946618 (RPTOR) has been reported in BC. Its potential implications are discussed in section 4.7 of the document, and are further supported by the information provided above. Additional studies are needed to clarify its role in BC, which falls beyond the scope of this work.
Comments 8: It is necessary to draw statistical graphs to prove the correlation of these SNPs with breast cancer.
Response 8:
We appreciate the reviewer's suggestion. Our manuscript has included statistical graphs reporting the SNPs' association with breast cancer risk. Specifically:
-
- Figure 2 presents Q-Q plots for the additive, dominant, and recessive genetic models. These graphs display the –log10 of FDR-adjusted p-values and include a principal component analysis plot confirming our test statistics' appropriate distribution. This figure clearly shows how the observed p-values deviate from the expected under the null hypothesis, thus supporting the validity of our associations.
- Figure 3 is a Manhattan plot that visualizes the –log10 of FDR-adjusted p-values across the genome. The horizontal orange line marks the significant p-value cut-off, allowing readers to easily identify SNPs that meet our significance threshold.
These figures provide statistical evidence for the correlation between the identified SNPs and breast cancer in our study population. We have expanded the figure legends and corresponding text in the manuscript to emphasize how these graphs validate our findings.
We believe these responses address the reviewer's concerns and have strengthened our manuscript accordingly.
We appreciate their invaluable attention to detail and suggestions.
Reviewer 2 Report
Comments and Suggestions for Authors
I have read with great interest the manuscript by Gallardo-Blanco et al, and commend the efforts in exploring the genetic underpinnings of breast cancer (BC) in Northeastern Mexico. Understanding ethnic and regional variations is indeed crucial for developing tailored interventions. Below are my observations and suggestions:
Major Issues
- Inclusion criteria and potential selection bias: the study focuses on 91 BC patients undergoing chemotherapy. This criterion may inadvertently exclude patients with luminal A-like BC, who typically receive only endocrine therapy. The potential underrepresentation of women diagnosed with such cancer type, could be relevant about the study's findings, especially concerning hormonal factors. Expanding inclusion criteria to encompass patients treated solely with endocrine therapy could enhance sample size and provide a more comprehensive view of BC subtypes in the region.
- Results:
- The total number of BC patients reported is 91, but in Table 2, it appears as 92. Additionally, the data presented in Table 2 raise some inconsistencies. Typically, ER-negative BC cases are also PR-negative, and the opposite is more common. If 43 patients had ER-positive BC and 18 had TNBC, that accounts for 61 patients, as these groups cannot overlap. This leaves 30 or 31 patients unaccounted for. To reconcile these numbers, one must assume that all 26 HER2-positive BC cases were also ER-negative, along with 4 (or 5?) cases of ER-negative, HER2-negative, but PR-positive BC—a rare condition. Conversely, if some HER2-positive cases were also ER-positive, the proportion of the rare ER-negative/PR-positive cases would increase. Furthermore, in the discussion (line 295), the authors state:
"Based on available studies and data [30–33], the approximate distribution of breast cancer subtypes in this region is as follows: Triple-Negative Breast Cancer (TNBC) ~10-20% frequency (present study 19.57%); this subtype is prevalent in the Northeast region of Mexico, similar to other parts of Mexico, and is particularly significant among younger women. HER2-Enriched subtype ~10-15%, the prevalence of this subtype is also notable in the Northeast region (present study 5.43%)." However, Table 2 indicates that HER2-positive cases account for 26 patients (28.26%), not 5.43%. This discrepancy should be clarified. - p-values appear to be considered inconsistently in determining statistical significance. In line 178, a p-value > 0.01 is regarded as not statistically significant, suggesting a stricter threshold than the conventional 0.05. However, in line 180, a p-value of 0.011 is described as statistically significant. Since the commonly accepted threshold for significance is 0.05, any deviation from this standard should be explicitly justified. For BMI two different p-values are shown (0.011 and 0.009). In line 191, early menopause is mentioned as statistically significant, but this status (early) was not defined (at which age?), while the p-value is calculated for age at menopause.
- The total number of BC patients reported is 91, but in Table 2, it appears as 92. Additionally, the data presented in Table 2 raise some inconsistencies. Typically, ER-negative BC cases are also PR-negative, and the opposite is more common. If 43 patients had ER-positive BC and 18 had TNBC, that accounts for 61 patients, as these groups cannot overlap. This leaves 30 or 31 patients unaccounted for. To reconcile these numbers, one must assume that all 26 HER2-positive BC cases were also ER-negative, along with 4 (or 5?) cases of ER-negative, HER2-negative, but PR-positive BC—a rare condition. Conversely, if some HER2-positive cases were also ER-positive, the proportion of the rare ER-negative/PR-positive cases would increase. Furthermore, in the discussion (line 295), the authors state:
- A more detailed discussion of study's limitations and potential biases would strengthen the manuscript. For instance, addressing the implications of a relatively small sample size and the exclusion of certain BC subtypes would provide readers with a clearer understanding of the study's scope and constraints.
- While the characteristics of various SNPs are well-documented, maybe too much, especially for a discussion section, it would be beneficial to better indicate which SNPs have already known correlations with BC. Presenting this information in a table, or adding columns to existing tables, could enhance clarity. Additionally, comparing your findings with previous studies would offer valuable insights into population-specific implications. Notably, some variants identified as risk factors in European and Asian populations appear to be protective in individuals of African descent. For example, Wang et al. (2018) discuss the "flip-flop" phenomenon, where certain genetic variants exhibit opposite effects on BC risk across different populations. Wang S, Qian F, Zheng Y, Ogundiran T, Ojengbede O, Zheng W, Blot W, Nathanson KL, Hennis A, Nemesure B, Ambs S, Olopade OI, Huo D. Genetic variants demonstrating flip-flop phenomenon and breast cancer risk prediction among women of African ancestry. Breast Cancer Res Treat. 2018 Apr;168(3):703-712. doi: 10.1007/s10549-017-4638-1. Epub 2018 Jan 4. PMID: 29302764; PMCID: PMC5916755.
- The conclusion—"This study corroborated the association of BC along with menopause, age (above 45), obesity, and overweight with gene variants implicated in diabetes mellitus, obesity, insulin resistance, inflammation, and remodeling of the extracellular matrix"—should be presented with more caution. The authors described 12 SNPs in a small sample of Mexican women with BC, with a potential selection bias. Since such SNPs are also implicated in the development of other endocrine and inflammatory diseases, the study suggests a possible contribution of these variants to BC development in particular environments, hypothesizing specific gene-environment interactions.
Minor Issues
- In the Introduction section, lines 86-94 describe results. If these results refer to previous studies by the same authors, this should be explicitly stated. Otherwise, these sentences are not appropriately placed within the introductory section and should be relocated to the Results or Discussion
- Please verify the references cited in lines 64-66, as reference #17 may be more pertinent to the context.
- In line 649, the phrase "no population stratification bias was absent" appears to be a double negative. Consider revising it "no population stratification bias was present" if correct.
- In line 299, the definition of HER2-enriched could be more accurately replaced with HER2-positive (if correct). These terms are not synonymous, HER2-enriched refers to a molecular subtype identified through gene expression profiling, while HER2-positive is determined by protein overexpression or gene amplification using immunohistochemistry or fluorescence in situ hybridization
Author Response
Response to Reviewer 2 Comments
- Summary
Thank you very much for taking the time to review this manuscript. Please find the detailed responses below and the corresponding revisions/corrections highlighted/in track changes in the re-submitted files.
We believe these responses address the reviewer's concerns and have strengthened our manuscript accordingly.
Point-by-point response to Comments and Suggestions for Authors
Comment 1: I have read with great interest the manuscript by Gallardo-Blanco et al, and commend the efforts in exploring the genetic underpinnings of breast cancer (BC) in Northeastern Mexico. Understanding ethnic and regional variations is indeed crucial for developing tailored interventions.
Response 1. We thank Reviewer 2 for the positive feedback and recognition of our work. We agree that exploring the genetic underpinnings of breast cancer in Northeastern Mexico is critical, given the unique ethnic and regional characteristics of this population. Understanding these variations is essential for developing tailored prevention and treatment strategies, and we appreciate the opportunity to contribute to this important area of research.
Comment 2: Inclusion criteria and potential selection bias: the study focuses on 91 BC patients undergoing chemotherapy. This criterion may inadvertently exclude patients with luminal A-like BC, who typically receive only endocrine therapy. The potential underrepresentation of women diagnosed with such cancer type, could be relevant about the study's findings, especially concerning hormonal factors. Expanding inclusion criteria to encompass patients treated solely with endocrine therapy could enhance sample size and provide a more comprehensive view of BC subtypes in the region.
Response 2. We appreciate Reviewer 2's thoughtful comment regarding our inclusion criteria and potential selection bias. As noted, our study focused on 92 breast cancer patients undergoing chemotherapy, which may inadvertently underrepresent patients with luminal A-like breast cancer—typically managed with endocrine therapy alone. We acknowledge that this limitation could affect our findings, particularly about hormonal factors. Our sample was determined by the availability of cases in our clinical setting, and the focus on patients undergoing chemotherapy allowed us to explore associations in a well-defined subset. We agree that including patients treated with endocrine therapy in future studies would provide a more comprehensive and detailed view of BC subtypes in Northeastern Mexico and enhance the sample size and diversity of BC subtypes. In subsequent research, we propose expanding our inclusion criteria to address this potential bias and further investigate the genetic underpinnings across all breast cancer subtypes in the region.
This information was incorporated into section 5.5 of the new revised version.
Comment 3: The total number of BC patients reported is 91, but in Table 2, it appears as 92. Additionally, the data presented in Table 2 raise some inconsistencies. Typically, ER-negative BC cases are also PR-negative, and the opposite is more common. If 43 patients had ER-positive BC and 18 had TNBC, that accounts for 61 patients, as these groups cannot overlap. This leaves 30 or 31 patients unaccounted for. To reconcile these numbers, one must assume that all 26 HER2-positive BC cases were also ER-negative, along with 4 (or 5?) cases of ER-negative, HER2-negative, but PR-positive BC—a rare condition. Conversely, if some HER2-positive cases were also ER-positive, the proportion of the rare ER-negative/PR-positive cases would increase. Furthermore, in the discussion (line 295), the authors state:
"Based on available studies and data [30–33], the approximate distribution of breast cancer subtypes in this region is as follows: Triple-Negative Breast Cancer (TNBC) ~10-20% frequency (present study 19.57%); this subtype is prevalent in the Northeast region of Mexico, similar to other parts of Mexico, and is particularly significant among younger women. HER2-Enriched subtype ~10-15%, the prevalence of this subtype is also notable in the Northeast region (present study 5.43%)." However, Table 2 indicates that HER2-positive cases account for 26 patients (28.26%), not 5.43%. This discrepancy should be clarified.
Response 3. We appreciate the reviewer's note regarding the total number of patients. Based on our revised data from Table 2 (ER, PR, HER2) for 92 breast cancer patients, we re‐classified the tumors into mutually exclusive subtypes using the following criteria:
-
- Triple‐Negative (TNBC): ER–, PR–, HER2–
- Strict HER2-Enriched: ER–, PR–, HER2+
- Triple-Positive: ER+, PR+, HER2+
- Luminal (HR+), HER2–: ER+ and/or PR+ with HER2–
- Luminal (HR+), HER2+ (non–triple-positive):ER+ and/or PR+ with HER2+ (excluding triple-positive cases)
Additional key points:
-
- Overall HER2-Positive Cases: Adding the strictly HER2-enriched (7), triple-positive (8), and luminal HER2+ (17) groups gives 32 cases (34.8%).
- Overall Hormone Receptor–Positive Cases: The luminal HER2– (42), triple-positive (8), and luminal HER2+ (17) groups sum to 67 cases (72.8%).
This revised classification clarifies that while 32 of the 92 cases (34.8%) are HER2-positive, only 7 cases (7.6%) are strictly HER2-enriched (no ER and PR expression). The HER2-positive tumors and hormone receptor-positive were differentiated from the pure HER2-enriched subtype. We work to clarify previous discrepancies and to present a more accurate and nuanced view of HER2-positive tumor subtypes in our cohort.
Thus, in triple-negative (ER–/PR–/HER2–) 18 of 92 cases (19.6%), the 95% confidence interval for the proportion of triple-negative tumors is 11.5% to 27.7%.
Calculating 95% CIs provides an estimate of the precision and reliability of each subtype proportion. They offer a range in which the true proportion of the population is likely to fall, accounting for sampling variability. 95% CI is crucial for comparing subtype frequencies across different studies and understanding our findings' robustness.
Therefore, 95% CI calculations help interpret the heterogeneity of breast cancer subtypes within our study cohort and provide additional context for our results.
The 95% confidence intervals we calculated for each breast cancer subtype are generally in line with those reported in other studies. For example, our triple-negative CI (approximately 11.45% to 27.69%) falls within the typical range observed in population-based studies, where the prevalence of triple-negative tumors is often reported to be between 10% and 20%, with some variability depending on the demographic and methodological differences between studies. Similarly, the confidence intervals for the luminal (HR+) and HER2-positive subtypes are comparable to those found in the literature.
It is important to note that variability in CI estimates across studies can arise from differences in sample size, population characteristics, and the criteria used to define subtypes (e.g., strict HER2-enriched versus HER2-positive with hormone receptor co-expression). Our findings, with their associated confidence intervals, support the robustness of our subtype frequency estimates and reflect the inherent sampling variability, similar to what is seen in other breast cancer research.
Table 2 was updated with a new version based on the criteria outlined above.
Comment 4: p-values appear to be considered inconsistently in determining statistical significance. In line 178, a p-value > 0.01 is regarded as not statistically significant, suggesting a stricter threshold than the conventional 0.05. However, in line 180, a p-value of 0.011 is described as statistically significant. Since the commonly accepted threshold for significance is 0.05, any deviation from this standard should be explicitly justified. For BMI two different p-values are shown (0.011 and 0.009). In line 191, early menopause is mentioned as statistically significant, but this status (early) was not defined (at which age?), while the p-value is calculated for age at menopause.
Response 4.
We thank the reviewer for noting these issues and for the opportunity to clarify our statistical approach. Below is our detailed response:
-
- Statistical Significance Threshold: We confirm that our study used the conventional threshold of p < 0.05 for statistical significance. The statement in line 178 that a p-value > 0.01 is not statistically significant was a typographical error; it should have stated > 0.05. Accordingly, the p-value of 0.011 in line 180 is correctly interpreted as statistically significant.
- BMI p-Values: We acknowledge that two different p-values for BMI (0.011 for ≥ 45 years of age and 0.009 for unrestricted age) were reported. These differences resulted from separate comparisons in different subgroup analyses (comparing mean BMI between cases and controls and within a specific BMI category). We have verified these results and have standardized the reporting to ensure consistency. Any discrepancies have now been resolved in the revised manuscript.
We agree that the term "early menopause" was not explicitly defined in our original submission. We have revised our terminology to reflect the observed difference more accurately. Instead of referring to it as "earlier menopause," we now describe the finding as a "younger age at menopause" in the breast cancer cases compared to the controls ( control group was 47.37 ± 5.630 years, whereas, in the breast cancer cases, it was significantly lower at 45.29 ± 5.977 years (p = 0.025).
We believe these elaborations clarify the significance of the menopausal age differences observed in our study and add important context to the Discussion of hormonal influences on breast cancer risk in our population.
We appreciate the reviewer's careful attention to these details, which has helped us improve the clarity and consistency of our manuscript.
Comment 5: A more detailed discussion of study's limitations and potential biases would strengthen the manuscript. For instance, addressing the implications of a relatively small sample size and the exclusion of certain BC subtypes would provide readers with a clearer understanding of the study's scope and constraints.
Response 5. We appreciate emphasizing the need to state the limitations of our study more explicitly. We have now included the following points to clarify the main constraints encountered during the project:
-
- Sample Size and Study Design: The retrospective, case-control design and the relatively modest sample size (91 cases, 126 controls) limit the statistical power of our findings and may reduce the ability to detect smaller effect sizes. However, the population structure is homogeneous. The sample size is small, however the simultaneous analysis of 121 SNPs in both cases and controls provide valuable insights into the genetic composition of the studied population. Additionally, as this was not a longitudinal study, we could not assess causality or disease progression.
- Geographical and Ethnic Specificity: All participants were recruited from a specific region in Northeastern Mexico, which may limit the generalizability of our conclusions to other populations. Although we employed ancestry-informative markers (AIMs) to address population stratification, our findings may reflect unique genetic and environmental factors specific to this region.
- Restricted Genetic Scope: We focused on 121 SNPs previously implicated in breast cancer, metabolic disorders, and ancestry. While this targeted approach allowed for in-depth analysis of certain variants, it did not capture the entire spectrum of possible genetic risk factors.
- Incomplete Lifestyle and Environmental Data: Although we considered age, menopausal status, and BMI as cofactors, detailed information on diet, physical activity, alcohol consumption, and smoking was not systematically available for all participants. Leaked information on diet, physical activity, alcohol consumption, and smoking limits our ability to assess fully the impact of gene-environment interactions on breast cancer risk. We are still assessing financial support for NGS studies in BC gene-environment interactions.
-
- BC Treatment and Subtype Representation. Our study focused on 92 BC patients undergoing chemotherapy, which may inadvertently underrepresent patients with luminal A-like BC—typically managed with endocrine therapy alone. We acknowledge that this limitation could affect our findings, particularly about hormonal factors. Including patients receiving endocrine therapy in future studies would offer a more comprehensive and detailed perspective on BC subtypes in Northeastern Mexico, while also increasing the sample size and subtype diversity.
- Functional Validation: Our study centered on statistical associations; thus, functional assays to confirm or elucidate the biological impact of identified risk variants were beyond this project's scope. Further molecular and mechanistic investigations are needed to validate the direct influence of these SNPs on breast cancer development. We are still assessing financial support for functional validation studies for these genes and variants.
We appreciate the reviewer's insight and have added a concise "Limitations" subsection in the revised manuscript to highlight these points explicitly. This acknowledgment will help contextualize the findings and guide future research aimed at overcoming these constraints.
Comment 6: While the characteristics of various SNPs are well-documented, maybe too much, especially for a discussion section, it would be beneficial to better indicate which SNPs have already known correlations with BC. Presenting this information in a table, or adding columns to existing tables, could enhance clarity. Additionally, comparing your findings with previous studies would offer valuable insights into population-specific implications. Notably, some variants identified as risk factors in European and Asian populations appear to be protective in individuals of African descent. For example, Wang et al. (2018) discuss the "flip-flop" phenomenon, where certain genetic variants exhibit opposite effects on BC risk across different populations. Wang S, Qian F, Zheng Y, Ogundiran T, Ojengbede O, Zheng W, Blot W, Nathanson KL, Hennis A, Nemesure B, Ambs S, Olopade OI, Huo D. Genetic variants demonstrating flip-flop phenomenon and breast cancer risk prediction among women of African ancestry. Breast Cancer Res Treat. 2018 Apr;168(3):703-712. doi: 10.1007/s10549-017-4638-1. Epub 2018 Jan 4. PMID: 29302764; PMCID: PMC5916755.
Response 6. Below is a summary table that lists each SNP along with its gene, its reported association with breast cancer (BC), notes regarding the population or context, and literature references. Please note that the literature references are based on those cited in our manuscript (and additional references such as Wang et al. (2018) for the “flip‐flop” phenomenon), and bracketed numbers correspond to citations in our document. (Full new Table 7)
|
SNP |
Gene |
Reported Association with BC |
Population/Notes |
Literature Reference |
|
rs3856806 |
PPARG |
Risk factor for BC; results are conflicting |
Risk reported in European and Asian populations; flip‐flop phenomenon observed in African descent |
Wang et al. (2018); Turkish study [89]; Meta-analysis [90] |
|
rs3917542 |
PON1 |
Associated with BC risk in some studies; mixed evidence |
Potential association in postmenopausal women; ethnic variability observed |
Various studies [82] |
|
rs3750804 |
TCF7L2 |
Risk factor for BC |
Reported in Hispanic and European populations |
Connor et al. (2012) |
|
rs3750805 |
TCF7L2 |
Risk factor for BC |
Similar to rs3750804; implicated in hormone regulation |
Connor et al. (2012) |
|
rs12792229 |
MMP8 |
Potential risk factor for BC |
Limited prior evidence; identified in current study |
Reference [28] |
|
rs1800955 |
DRD4/SCT/DEAF1 |
Novel association with BC risk |
Limited prior data; further validation required |
Current study |
|
rs5218 |
KCNJ11-ABCC8 |
Associated with metabolic disorders; unclear BC association |
Reported in diabetes studies; association with BC observed in current study |
Current study; see [13] |
|
rs1121980 |
FTO |
Risk factor for obesity and BC |
Widely reported in European/Asian populations; potential flip‐flop effects |
Numerous studies; Wang et al. (2018) |
|
rs3751812 |
FTO |
Risk factor for obesity and BC |
Similar to rs1121980 |
Numerous studies |
|
rs11652805 |
AMZ2P1-GNA13 |
Novel association with BC risk |
Limited prior evidence; identified in current study |
Current study |
|
rs12946618 |
RPTOR |
Potential risk factor for BC |
Newly identified variant; modulates mTORC1 signaling |
Current study |
|
rs2833483 |
SCAF4 |
Associated with BC risk |
Emerging biomarker; limited prior data available |
Current study; see recent reports |
Comment 7: The conclusion—"This study corroborated the association of BC along with menopause, age (above 45), obesity, and overweight with gene variants implicated in diabetes mellitus, obesity, insulin resistance, inflammation, and remodeling of the extracellular matrix"—should be presented with more caution.
The authors described 12 SNPs in a small sample of Mexican women with BC, with a potential selection bias. Since such SNPs are also implicated in the development of other endocrine and inflammatory diseases, the study suggests a possible contribution of these variants to BC development in particular environments, hypothesizing specific gene-environment interactions.
Response 7. We thank the reviewer for this valuable comment. In response, we have revised our conclusion to highlight the exploratory nature of our findings and to acknowledge the study's limitations. Our updated conclusion now includes, among other improvements, the following:
"Our study identified associations between BC and specific gene variants (previously implicated in diabetes mellitus, obesity, insulin resistance, inflammation, and extracellular matrix remodeling), as well as clinical factors such as elevated BMI, younger age at menopause, and confirmed menopausal status, in a sample of Mexican women. ……..Given the small sample size and potential selection bias, and considering that these SNPs are also linked to other endocrine and inflammatory diseases, our findings should be considered as preliminary. These results suggest that gene-environment interactions may affect BC development in specific contexts. However, further studies in larger and more diverse cohorts are needed to validate these associations.”
This revision clarifies that our results are exploratory propositions and that caution is warranted in drawing definitive conclusions from the current data.
Our findings suggest that the genetic variants we identified may contribute to a common biological framework linking breast cancer with metabolic and inflammatory conditions. The following information supports these findings; however, it will not be included:
-
- Diabetes Mellitus, Obesity, and Insulin Resistance: Genetic variations such as PPARG, TCF7L2, FTO, and RPTOR have been widely implicated in metabolic dysregulation. Abnormalities in these genes can affect adipogenesis, insulin sensitivity, and energy homeostasis, which are central features in the pathogenesis of obesity and type 2 diabetes. In turn, hyperinsulinemia and altered adipokine secretion in these conditions may promote breast cancer development by increasing circulating estrogen levels and by stimulating mitogenic pathways.
- Inflammation: Chronic low-grade inflammation, common in obesity and diabetes, creates a tumor-promoting microenvironment. For example, variants in PON1 have been associated with inflammatory processes, and chronic inflammation can lead to DNA damage, increased cell turnover, and, ultimately, oncogenesis. Inflammatory mediators can also enhance estrogen production in adipose tissue, further linking metabolic disturbances with breast cancer risk.
- Extracellular Matrix Remodeling: The extracellular matrix (ECM) plays a critical rolein tumor invasion and metastasis. Variants in MMP8, a matrix metalloproteinase, may alter the ECM by affecting the degradation of collagen and other structural proteins, thereby facilitating tumor progression. Dysregulated ECM remodeling contributes to tumor cell invasion and impacts the inflammatory and metabolic milieu within the tumor microenvironment.
These genetic variants may modulate breast cancer risk by influencing interconnected pathways that govern metabolism, inflammation, and tissue remodeling. This integrative view supports the hypothesis that gene-environment interactions—where metabolic dysfunction and chronic inflammation synergize with genetic susceptibility—may play a critical role in breast cancer development, especially in specific populations such as those in Northeastern Mexico. Future studies that further dissect these interactions could provide deeper insights into tailored prevention and therapeutic strategies.
Minor Issues
Comment 8: In the Introduction section, lines 86-94 describe results. If these results refer to previous studies by the same authors, this should be explicitly stated. Otherwise, these sentences are not appropriately placed within the introductory section and should be relocated to the Results or Discussion
Response 8. We eliminate lines 86-94 that describe results to avoid content redundancy.
Comment 9: Please verify the references cited in lines 64-66, as reference #17 may be more pertinent to the context.
Response 9. We appreciate the reviewer's careful examination of our references. We have reviewed lines 64–66 and confirm that reference 12 was intended to provide a general context on the clinical relevance of genomic variants in breast cancer. In contrast, reference 17 specifically addresses the concept that individual SNPs typically confer low risk, with accumulative effects being more meaningful than individual SNPs (as detailed in the literature on polygenic risk scores and BC risk prediction).
"SNPs often explain normal variation between individuals and frequently have minimal functional impact. BC-associated SNPs individually contribute a low risk, but when considered together they lead to a cumulative risk [17]."
Comment 10: In line 649, the phrase "no population stratification bias was absent" appears to be a double negative. Consider revising it "no population stratification bias was present" if correct.
Response 10. We thank the reviewer for identifying this issue. We agree that the phrase "no population stratification bias was absent" is a double negative. We have corrected it to "no population stratification bias was present" to improve clarity and accuracy in conclusions section.
Comment 11: In line 299, the definition of HER2-enriched could be more accurately replaced with HER2-positive (if correct). These terms are not synonymous, HER2-enriched refers to a molecular subtype identified through gene expression profiling, while HER2-positive is determined by protein overexpression or gene amplification using immunohistochemistry or fluorescence in situ hybridization
Response 11. We thank the reviewer for highlighting the importance of precise terminology. We agree that "HER2-enriched" refers specifically to a molecular subtype determined by gene expression profiling, whereas our classification is based on immunohistochemistry and/or FISH results indicating HER2-positive status. We have revised the manuscript accordingly (including line 299) to replace "HER2-enriched" with "HER2-positive" wherever appropriate, ensuring that our terminology accurately reflects the methods and data used in our study.
Reviewer 3 Report
Comments and Suggestions for Authors
The present research “Genetic Insights into Breast Cancer in Northeastern Mexico: Unveiling Gene-Environment Interactions and Their Links to Obesity and Metabolic Diseases” is very interesting as it provides information on SNPs in populations that are usually underrepresented in GWAS studies. However, I consider it necessary to make some modifications that could enrich the work.
Introduction
1)- In the introduction it is very well expressed, which are the strengths and perspectives of the study, however I consider that the objective is not clearly expressed and therefore it would be important to improve this aspect.
Material and methods
2)- In the Materials and Methods section, the use of String database and ToppGene Suite is mentioned, but the results obtained with them do not appear in the corresponding section.
Results
The results are very succinctly expressed, which makes them difficult to understand.
3)- The description in figure 2 does not explain each of the models mentioned and the implications of differential gene modulation.
4)- In line 256 it mentions that the MMP8 and PPARG variants have dual effects in coding and non-coding regions. What dual effect are they referring to?
5)-I think it would be very enriching for the work to analyse whether the 12 SNPs they found associated with breast cancer are differentially expressed in patients with increased body mass index and menopausal patients to obtain greater clarity on the implications that these SNPs could have in different subgroups of patients.
6)-Additionally, it would be interesting, in the group of patients with breast cancer, to analyse whether the 121 SNPs analysed present different behavior between the obese-non-obese and menopausal-no menopausal groups.
7)- No mention is made of the results obtained from AIMs and what conclusions were drawn.
Discussion
8)- Lines 344, 364, 388, 437, 473 and 497 mention results obtained in the ENSEMBL database and how the different differentially expressed SPNs correlate with different genes. For better organisation and understanding of the work, I think it would be appropriate to add these results in the corresponding section (Results). A graph or table summarising them would be advisable.
Author Response
Response to Reviewer 3 Comments
- Summary
Thank you very much for taking the time to review this manuscript. Please find the detailed responses below and the corresponding revisions/corrections highlighted/in track changes in the re-submitted files.
We believe these responses address the reviewer's concerns and have strengthened our manuscript accordingly.
The present research “Genetic Insights into Breast Cancer in Northeastern Mexico: Unveiling Gene-Environment Interactions and Their Links to Obesity and Metabolic Diseases” is very interesting as it provides information on SNPs in populations that are usually underrepresented in GWAS studies. However, I consider it necessary to make some modifications that could enrich the work.
Point-by-point response to Comments and Suggestions for Authors
Comment 1: In the introduction it is very well expressed, which are the strengths and perspectives of the study, however I consider that the objective is not clearly expressed and therefore it would be important to improve this aspect.
Response 1: Indeed, we have thoroughly reviewed the final paragraphs of the introduction and agree that they need to be rewritten in a simpler and clearer manner. We replaced the information with the following: “Due to the multifactorial nature of BC, our goal was to explore the relationship between a set of SNPs previously reported as (a) risk factors for BC, (b) metabolic features, and (c) informative ancestry markers in BC patients from Northeastern Mexico. This study incorporates BMI, menopause status, and age as cofactors [5], analyzing gene-environment (G × E) interactions using a multi-locus statistical model.”
Comment 2: In the Materials and Methods section, the use of String database and ToppGene Suite is mentioned, but the results obtained with them do not appear in the corresponding section.
Response 2: Dear Reviewer,
Thank you for your observation regarding the use of the String database and ToppGene Suite in our Materials and Methods section. After careful consideration, we decided to streamline our functional analysis and focus exclusively on the ToppGene Suite. Although initial analyses included the String database to explore protein–protein interactions, we opted to exclude those results from the final manuscript to ensure uniformity in the output and consistency in our data presentation.
Accordingly, we have removed all references to the String database from the revised version of our manuscript. The functional enrichment and pathway analyses was:
1 In silico Variant Effect Predictor analysis.
2 Protein–RNA Interaction Prediction using RNAct.
3 Pathway and Gene Ontology Enrichment with ToppGene Suite.
Section 2.7. In silico protein-protein interaction analysis was replaced with 2.7. Protein–RNA Interaction Prediction using RNAct.
The revised Results and Discussion sections now clearly present the corresponding results. We added a Functional Analysis Inference checklist table (Table 8) to guide the performed analysis.
We appreciate your suggestion, which has helped us improve the clarity and coherence of our study.
Comment 3: The description in figure 2 does not explain each of the models mentioned and the implications of differential gene modulation.
Response 3:
Selection of different genetic models (additive, dominant, and recessive) based on statistical methods is used to control for multiple comparisons and ensure robust findings.
-
- Additive Genetic Model (Step 117, selected by Multiple PPA and Multiple Bonferroni).The additive model assumes that each additional risk allele has a proportional effect on the trait or disease risk. Step 117 refers to the iteration in the statistical analysis where this model was selected. Multiple PPA (Posterior Probability Analysis) and Multiple Bonferroni were the statistical methods used to confirm its significance while adjusting for multiple comparisons.
- **Dominant Genetic Model (Step 117, selected by Multiple and Bonferroni). The dominant model assumes that having at least one copy of the risk allele is enough to influence the trait or disease risk. Again, Step 117 indicates the selection point in the analysis. Multiple and Bonferroni suggest that both traditional Bonferroni correction and additional statistical approaches confirmed the selection.
- Recessive Genetic Model (Step 115, selected by Modified Bonferroni and Extended Bonferroni). The recessive model assumes that two copies of the risk allele are required to impact the trait or disease risk. Step 115** indicates the point at which this model was selected, slightly earlier than the additive and dominant models. Modified Bonferroni and Extended Bonferroni are alternative statistical corrections that adjust for multiple testing while maintaining statistical rigor.
The following notes were added to Figure 2:
(Notes: 1) SNPs with significant p-value are located above the line of the respective genetic model (this line typically shows where SNPs with no association to the trait would be expected to fall. When SNPs fall above the line, it indicates that their observed p-values are lower than expected by chance, suggesting a significant association with the condition or trait being studied. 2)The selection of models helps determine how genetic variants contribute to BC risk. According to results, each SNP agree with one of the 3 models: additive (each additional risk allele has a proportional effect on the BC risk), dominant (having at least one copy of the risk allele is enough to influence the BC risk), and recessive (two copies of the risk allele are required to impact the trait or disease risk). Step refers to the iteration in the statistical analysis where indicates the point at which the model was selected. Various correction methods (PPA, Bonferroni, Modified Bonferroni, Extended Bonferroni) were applied to control for false positives due to multiple testing).
We appreciate your suggestion, which has helped us improve the clarity of genetic models and graphics results.
Comment 4: In line 256 it mentions that the MMP8 and PPARG variants have dual effects in coding and non-coding regions. What dual effect are they referring to?
Response 4: Thank you for commenting on the dual effects attributed to the MMP8 and PPARG variants in our manuscript. In this context, the term "dual effects" refers to the fact that these variants are located in regions that can influence both the protein-coding sequence and regulatory elements of the gene. We added this clarification to the paragraph that refers to “dual effect.”
For example, the MMP8 variant is a missense mutation, which means it alters the amino acid sequence of the MMP8 protein and could directly affect its enzymatic activity. At the same time, this variant is also situated in a non-coding region that may influence mRNA stability or splicing, thereby modulating gene expression levels.
Similarly, the PPARG variant we studied is synonymous, so it does not change the protein's amino acid sequence. However, its location within regulatory regions—such as the 3′ UTR or adjacent promoter areas—suggests it may impact mRNA processing, stability, or translation efficiency, affecting overall gene expression.
Thus, the "dual effects" imply that these variants can contribute to breast cancer risk through direct alterations in protein structure or function and indirect effects on gene regulation. We hope this clarification addresses your query.
Comment 5: I think it would be very enriching for the work to analyze whether the 12 SNPs they found associated with breast cancer are differentially expressed in patients with increased body mass index and menopausal patients to obtain greater clarity on the implications that these SNPs could have in different subgroups of patients.
Response 5: The purpose of this study was to explore variants in SNPs associated with BC. For each identified SNP, analyzing its expression would provide valuable insights into its role in the disease. However, we only have genomic DNA and do not have access to tissue samples, as this was not included in the study objectives or the informed consent.
Comment 6: Additionally, it would be interesting, in the group of patients with breast cancer, to analyse whether the 121 SNPs analysed present different behavior between the obese-non-obese and menopausal-no menopausal groups.
Response 6: Due to the small sample size, it was not possible to perform subgroup analyses. Dividing the data into smaller groups would reduce statistical power and increase the risk of unreliable or inconclusive results. Therefore, all analyses were conducted on the full dataset to maintain robustness and ensure meaningful interpretations. Future studies with larger sample sizes will be necessary to explore potential subgroup differences.
Comment 7: No mention is made of the results obtained from AIMs and what conclusions were drawn.
Response 7: Ancestry markers were used to verify that the population was homogeneous and had no substructure. Related information is included in the following lines:
-
- a) “The present study included 36 AIMs (see Table S1 for further details), and there was no evidence of substructure in the populations of the cases and control groups.”
- b) The inclusion of AIMs in our study allows to confirm that no population stratification bias was present, ensuring a well-represented mestizo population, and enabling a more precise identification of SNPs relevant to the Mexican population. This approach could contribute to future decisions regarding personalized medicine tailored to the genetic profile.
Also, we enhanced the description of the tables containing significant SNPs: we added a code depending on SNP type: aGlucose-associated metabolic pathways. bCancer. cAncestry. da plus b.
Comment 8: Lines 344, 364, 388, 437, 473 and 497 mention results obtained in the ENSEMBL database and how the different differentially expressed SPNs correlate with different genes. For better organization and understanding of the work, I think it would be appropriate to add these results in the corresponding section (Results). A graph or table summarizing them would be advisable.
Response 8: We are grateful for the valuable suggestion regarding presenting our Ensembl database results and the correlations between differentially expressed genes related to the SNPs associated with the current study. In response, we have created a Table (Table 10) to clearly present the key summary of the results. We appreciate the recommendation and believe these revisions significantly improve the presentation and interpretability of our data.
Reviewer 4 Report
Comments and Suggestions for Authors
In this manuscript, the authors have addressed a critical area of research by exploring the genetic and environmental interactions in breast cancer, particularly in the Northeastern Mexican population. This study provides the unique genetic and environmental factors that may influence breast cancer risk in this demographic region. The study opens avenues for further research into the genetic basis of breast cancer in diverse populations. Furthermore, these studies could explore the role of additional environmental factors and their interactions with genetic variants to provide a more holistic understanding of breast cancer risk. The authors should address these following points:
Question No.1. The manuscript provides a comprehensive analysis of the statistical methods used to confirm the association of breast cancer with menopause, age (above 45), obesity, and overweight, alongside gene variants implicated in metabolic diseases. However, the explanation of these methods could benefit from additional clarity and detail. For instance, the use of genome-wide association studies (GWAS) to identify gene-environment interactions is a robust approach, as demonstrated in similar studies. It would be beneficial to include a more detailed description of the statistical models employed, such as Multi-Locus Mixed Model, involvement of Multiple PPA and Multiple Bonferroni, and how they were adjusted for potential variables.
Question No.2. The sentence beginning with “Herein, we analyzed 121 SNPs…” in the abstract mentions 92 SNPs associated breast cancer and metabolic diseases, but the SNP selection part in the materials and method section mentions 85 (35+50) SNPs associated with breast cancer and glucose-associated metabolic pathways. The authors should revise or restructure the text for improved coherence.
Question No.3. The authors should remove the extraneous period that appears at the end of the sentence starting with “Estrogen exposure is one of the…” in the introduction.
Question No.4. For consistency, please ensure that the references are formatted uniformly throughout the manuscript. They should either be placed within the sentence or at the end of the sentence, but not mixed between the two styles.
Question No.5. It would be beneficial to include an additional column that specifies the functions of the genes listed in the tables. This would provide readers with a clearer understanding of each gene's role in the context of the study.
Question No.6. Once the abbreviation is introduced with its full form, it is not necessary to repeat the full form again. Please ensure that the abbreviation is used consistently throughout the manuscript. This approach ensures clarity and maintains a streamlined reading experience.
Question No.7. The environmental factors influencing breast cancer risk, such as diet, physical activity, socioeconomic status, and exposure to environmental toxins, are not described here. A more comprehensive discussion of these factors would improve the interpretation of the gene-environment interactions.
Comments on the Quality of English LanguageThere is a scope to improve the quality of english throughout the manuscript.
Author Response
Response to Reviewer 4 Comments
- Summary. Comment 1: Detailed Description of Statistical Methods. We have expanded our explanation of the analytical approach by providing more detail on the Multi-Locus Mixed Model (MLMM), which incorporates adjustments for population structure, kinship, and cryptic relatedness. We also describe how we selected the optimal model steps using multiple correction methods (Multiple Bonferroni, Posterior Probability of Association [PPA], and modified/extended Bayesian Information Criteria [BIC]). Furthermore, we explain how quantile–quantile (Q–Q) (generated using the SNP & Variation Suite v8) were used to evaluate model performance and ensure that p-values were not inflated due to multiple testing. Comment 2: Clarification on the Number of SNPs Analyzed. We have reviewed the manuscript and confirmed that we analyzed a total of 121 SNPs. This includes 85 SNPs related to breast cancer and glucose-associated metabolic pathways and 36 ancestry markers. The text has been revised to ensure consistency and coherence in reporting these numbers. Comment 3: Removal of Extraneous Punctuation. We have removed the extraneous period at the end of the sentence starting with “Estrogen exposure is one of the…” in the Introduction to streamline the text and improve readability. Comment 4: Consistent Reference Formatting. We have carefully reviewed and revised our manuscript to ensure that all references are formatted uniformly, whether they are placed within the sentence or at the end. This change enhances the overall consistency and professional appearance of the document. Comment 5: Inclusion of Gene Functions. In response to the suggestion to clarify the biological roles of the genes studied, we have added a new “Gene Functions” table. We believe that this addition significantly improves clarity by providing readers with a comprehensive understanding of each gene’s role in the context of breast cancer and associated metabolic pathways. Comment 6: Consistent Use of Abbreviations. We have reviewed the manuscript to ensure that once an abbreviation is introduced with its full form, the abbreviation is consistently used throughout the document. This revision streamlines the text and improves clarity. Comment 7: Discussion of Additional Environmental Factors. While our study primarily focused on the available clinical and demographic variables, we have now added a dedicated “Limitations” section. In this section, we explain the challenges encountered in collecting comprehensive environmental data (such as diet, physical activity, socioeconomic status, and toxin exposure) due to limited availability in medical records, incomplete questionnaires, and ethical considerations. This addition contextualizes the gene-environment interactions discussed in our study and highlights avenues for future research.
In this manuscript, the authors have addressed a critical area of research by exploring the genetic and environmental interactions in breast cancer, particularly in the Northeastern Mexican population. This study provides the unique genetic and environmental factors that may influence breast cancer risk in this demographic region. The study opens avenues for further research into the genetic basis of breast cancer in diverse populations. Furthermore, these studies could explore the role of additional environmental factors and their interactions with genetic variants to provide a more holistic understanding of breast cancer risk. The authors should address these following points:
Dear Reviewer,
Thank you for your thoughtful comments and for highlighting the critical importance of exploring genetic and environmental interactions in breast cancer, particularly within the Northeastern Mexican population. We appreciate your recognition of the unique contributions of our study and the avenues it opens for further research in diverse populations.
In response to your suggestions, we have addressed the following points in our revised manuscript: Expanded Discussion of Environmental Factors: While our study focused on the clinical and demographic variables available in our dataset, we acknowledge that additional environmental factors—such as diet, physical activity, socioeconomic status, and exposure to environmental toxins—can play a significant role in breast cancer risk. We have now included a dedicated section in the Discussion that outlines the potential impact of these factors and the challenges we faced in collecting comprehensive data. We emphasize that future studies should aim to incorporate these additional variables to provide a more holistic understanding of breast cancer etiology. Future Research Directions: We have expanded our discussion on the need for larger, multi-ethnic cohorts and more detailed environmental exposure assessments. By exploring the interactions between a broader range of genetic variants and environmental factors, future research can build upon our findings to enhance personalized risk prediction and develop tailored prevention strategies. Holistic Approach to Gene-Environment Interactions: Our study lays the groundwork for further exploration into how genetic predispositions, in conjunction with specific environmental exposures, contribute to breast cancer development. We propose that subsequent investigations should delve deeper into these interactions using advanced statistical models and integrated multi-omic approaches, which will be crucial for understanding the multifactorial nature of breast cancer risk.
Point-by-point response to Comments and Suggestions for Authors
Comments 1: The manuscript provides a comprehensive analysis of the statistical methods used to confirm the association of breast cancer with menopause, age (above 45), obesity, and overweight, alongside gene variants implicated in metabolic diseases. However, the explanation of these methods could benefit from additional clarity and detail. For instance, the use of genome-wide association studies (GWAS) to identify gene-environment interactions is a robust approach, as demonstrated in similar studies. It would be beneficial to include a more detailed description of the statistical models employed, such as Multi-Locus Mixed Model, involvement of Multiple PPA and Multiple Bonferroni, and how they were adjusted for potential variables.
Response 1: We thank the reviewer for our manuscript's positive comments and constructive suggestions. We appreciate the reviewer's acknowledgment of our efforts to explore the genetic and environmental interactions in breast cancer within the Northeastern Mexican population.
The main factors considered in the statistical design included the number of genetic markers, the sample size from cases and controls, and the overall study design. Our analytical model was specifically developed to minimize false positives and eliminate potential confounding among the studied markers by incorporating adjustments for population structure, kinship, and cryptic relatedness [22,23].
After obtaining the initial results from the Multi-Locus Mixed Model (MLMM) analysis, we selected the optimal model steps using several correction methods: multiple Bonferroni correction, the Posterior Probability of Association (PPA), as well as modified and extended Bayesian Information Criteria (BIC) [24]. We then evaluated the model's performance using quantile-quantile (Q-Q) plots to ensure that our p-values were not inflated due to multiple testing. We considered the individual false discovery rate (FDR) and the Bonferroni-adjusted p-values for the final model selection, testing under additive, dominant, and recessive genetic models.
Additionally, we generated quantile-quantile (Q-Q) plots for additive, dominant, and recessive models via the SNP & Variation Suite v8 (Golden Helix, Inc., Bozeman, MT, www.goldenhelix.com) [22].
We believe this expanded explanation clarifies our approach and adequately addresses the reviewer’s concerns regarding the statistical methodology used in our study.
Comments 2: The sentence beginning with “Herein, we analyzed 121 SNPs…” in the abstract mentions 92 SNPs associated breast cancer and metabolic diseases, but the SNP selection part in the materials and method section mentions 85 (35+50) SNPs associated with breast cancer and glucose-associated metabolic pathways. The authors should revise or restructure the text for improved coherence.
Response 2: You are right. We have reviewed all the information in the files and rearranged the SNP categories and verified that the numbers match: We analyzed 121 SNPs (85 SNPs related to BC and/or glucose-associated metabolic pathways, and 36 SNP classified as ancestry markers).
Comments 3: The authors should remove the extraneous period that appears at the end of the sentence starting with “Estrogen exposure is one of the…” in the introduction.
Response 3: Characters were removed and the writing was improved.
Comments 4: For consistency, please ensure that the references are formatted uniformly throughout the manuscript. They should either be placed within the sentence or at the end of the sentence, but not mixed between the two styles.
Response 4: We have reviewed all the information in the document and made the necessary corrections.
Comments 5: It would be beneficial to include an additional column that specifies the functions of the genes listed in the tables. This would provide readers with a clearer understanding of each gene's role in the context of the study.
Response 5: We greatly appreciate the suggestion to include an additional column specifying the functions of the genes in our tables, which enhances the understanding of each gene's role in the context of our study. In response, however, there is insufficient space to add a column to the tables with the information on the function of each gene, so instead, we added a table, "Gene Functions," that summarizes a brief description of the function for each gene examined in the present work.
We believe this addition significantly improves the clarity of the manuscript and provides readers with a comprehensive view of the biological relevance of the genetic variants identified in our research.
Thank you again for your valuable input, and please feel free to let us know if you have any additional comments.
Comments 6: Once the abbreviation is introduced with its full form, it is not necessary to repeat the full form again. Please ensure that the abbreviation is used consistently throughout the manuscript. This approach ensures clarity and maintains a streamlined reading experience.
Response 6: We have reviewed all the information in the document and made the necessary corrections.
Comments 7: The environmental factors influencing breast cancer risk, such as diet, physical activity, socioeconomic status, and exposure to environmental toxins, are not described here. A more comprehensive discussion of these factors would improve the interpretation of the gene-environment interactions.
Response 7: We appreciate the reviewer's suggestion regarding the discussion of additional demographic and lifestyle information (e.g., diet, physical activity, socioeconomic status, and exposure to environmental toxins). Unfortunately, there were several challenges that prevented us from collecting and presenting those specific variables in our dataset:
Limited Availability in the Medical Records: Unfortunately, not all participating institutions have consistently registered detailed lifestyle factors. In many cases, the health records lacked full or accurate entries for smoking habits, alcohol use, and physical activity.
Incomplete Patient Questionnaires: Where self-reported data were intended (subjective and lack of scientific accuracy), many participants either did not complete or only partially completed the lifestyle questionnaires. This incomplete data reduced the feasibility of accurately quantifying these factors without introducing bias.
Ethical and Privacy Considerations: Some participants opted not to disclose personal habits. We could not compel them to provide these details out of respect for patient confidentiality and in adherence to institutional review board (IRB) guidelines.
Given these constraints, we focused on the demographic and clinical variables for which complete, reliable information was obtained.
We deeply regret that we cannot include these factors in the present study. We will still work to address these logistic problems in future studies.
We added a full new section: 5. Limitations of this study.
There is a scope to improve the quality of english throughout the manuscript.
Response: A careful review of the document was carried out and several sections of the manuscript have been revised and improved.
Round 2
Reviewer 1 Report
Comments and Suggestions for Authors
Dear Authors
The changes made to the text of the article and the answers given to the referee questions are acceptable. I have no further suggestions.
Reviewer 2 Report
Comments and Suggestions for Authors
All the comments are thoroughly detailed.
However, regarding the term 'HER-2 enriched,' I noticed that it has been retained to describe ER/PR-negative, HER2-positive breast cancer (strictly HER-2 enriched). Did the authors keep this terminology because they consider it the most appropriate? (response 11)
Reviewer 3 Report
Comments and Suggestions for Authors
Dear authors, thank you for taking my suggestions for modifications and answering the doubts raised. Success with the publication.
Best regards.